# From Play to Replay: Composed Video Retrieval for Temporally Fine-Grained Videos

**Animesh Gupta**[1]    **Jay Parmar**[1]    **Ishan Rajendrakumar Dave**[2]    **Mubarak Shah**[1]
[1]Center for Research in Computer Vision, University of Central Florida    [2]Adobe

## Abstract

Composed Video Retrieval (CoVR) retrieves a target video given a query video and a modification text describing the intended change. Existing CoVR benchmarks emphasize appearance shifts or coarse event changes and therefore do not test the ability to capture subtle, fast-paced temporal differences. We introduce TF-CoVR, the first large-scale benchmark dedicated to temporally fine-grained CoVR. TF-CoVR focuses on gymnastics and diving, and provides 180K triplets drawn from FineGym and FineDiving datasets. Previous CoVR benchmarks, focusing on temporal aspect, link each query to a single target segment taken from the same video, limiting practical usefulness. In TF-CoVR, we instead construct each <query, modification> pair by prompting an LLM with the label differences between clips drawn from different videos; every pair is thus associated with multiple valid target videos (3.9 on average), reflecting real-world tasks such as sports-highlight generation. To model these temporal dynamics, we propose TF-CoVR-Base, a concise two-stage training framework: (i) pre-train a video encoder on fine-grained action classification to obtain temporally discriminative embeddings; (ii) align the composed query with candidate videos using contrastive learning. We conduct the first comprehensive study of image, video, and general multimodal embedding (GME) models on temporally fine-grained composed retrieval in both zero-shot and fine-tuning regimes. On TF-CoVR, TF-CoVR-Base improves zero-shot mAP@50 from 5.92 (LanguageBind) to 7.51, and after fine-tuning raises the state-of-the-art from 19.83 to 27.22. We have released our dataset and code publicly available at https://github.com/UCF-CRCV/TF-CoVR.

## 1   Introduction

Recent progress in content-based image retrieval has evolved into multimodal *composed image retrieval* (CoIR) [49, 1, 23], where a system receives a *query image* and a short *textual modification* and returns the image that satisfies the composition. *Composed video retrieval* (CoVR) [41] generalizes this idea, asking for a target video that realizes a user-described transformation of a query clip, for example, "same river landscape, but in springtime instead of autumn" (Fig. 1a) or "same pillow, but picking up rather than putting down"(Fig. 1b).

Existing CoVR benchmarks cover only a limited portion of the composition space. For example, WebVid-CoVR [41] (Fig. 1a) is dominated by appearance changes and demands minimal temporal reasoning, while Ego-CVR [9] restricts the query and target to different segments of a single video (Fig. 1b). In practice, many high-value applications depend on *fine-grained* motion differences: surgical monitoring of subtle patient movements [38], low-latency AR/VR gesture recognition [47], and sports analytics where distinguishing a 1.5-turn from a 2-turn somersault drives coaching feedback [8, 26]. The commercial impact is equally clear: the Olympic Broadcasting Service AI highlight pipeline in Paris 2024 increased viewer engagement 13 times in 14 sports [13]. No public dataset currently evaluates CoVR at this temporal resolution.

39th Conference on Neural Information Processing Systems (NeurIPS 2025) Track on Datasets and Benchmarks.

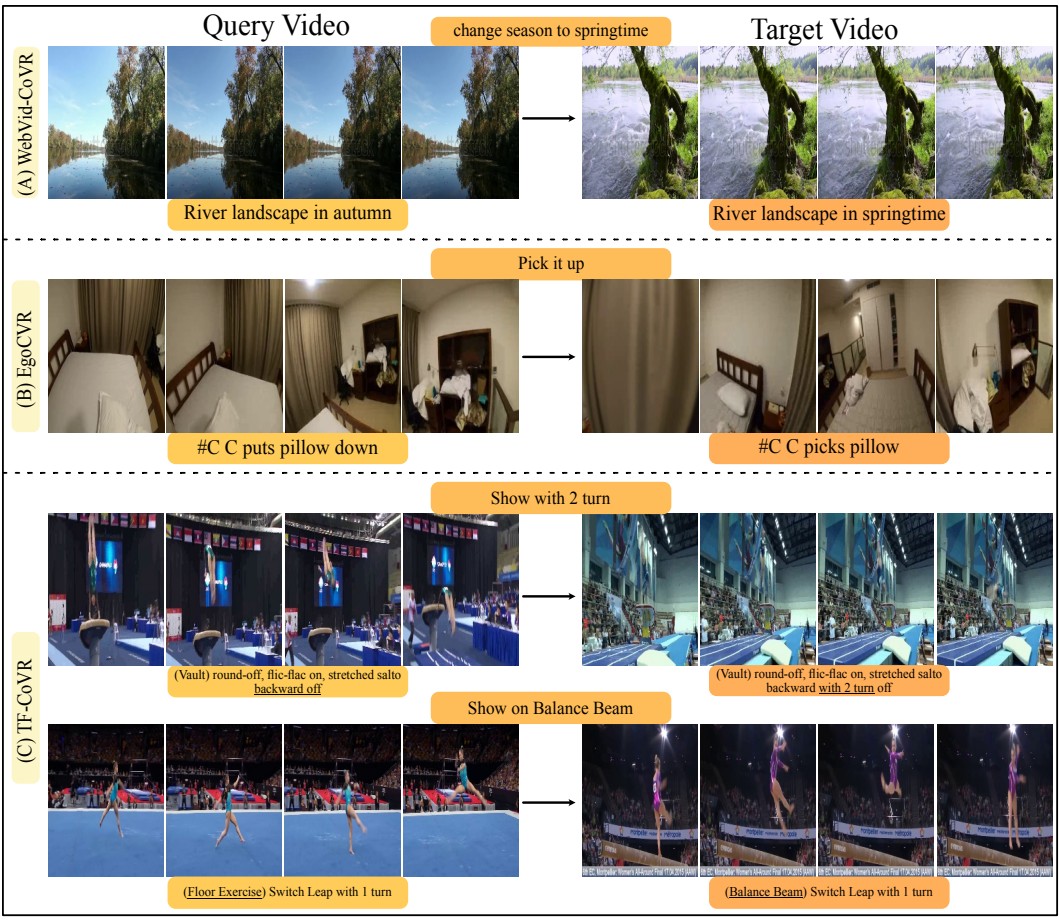

Figure 1: Comparison of composed-retrieval triplets in WebVid-CoVR, Ego-CVR, and TF-CoVR. (a) WebVid-CoVR targets appearance changes. (b) Ego-CVR selects the target clip from a different time-stamp of the *same* video, showing a new interaction with the same object. (c) TF-CoVR supports two fine-grained modification types: temporal change- varying sub-actions within the same event (row 3), and event change- the same sub-action performed on different apparatuses (row 4).

To address these limitations, we present *TF-CoVR* (T̲emporally F̲ine-grained C̲omposed V̲ideo R̲etrieval), a large-scale benchmark for composed retrieval in gymnastics and diving constructed from the temporally annotated FineGym [32] and FineDiving [46] datasets. Previous work such as Ego-CVR [9] restricts query and target clips to different segments of a *single* video; in practice, however, relevant results often come from distinct videos. TF-CoVR instead provides 180K triplets, each containing a query video, a textual modification, and one or more ground-truth target videos. We call each ⟨query, modification⟩ pair a *composed query*. The benchmark covers both event-level changes (e.g. the same sub-action on different apparatuses) and fine-grained sub-action transitions (e.g. varying rotation counts or entry/exit techniques), yielding a setting that reflects real-world temporally fine-grained retrieval far more closely than existing datasets. A thorough comparison with prior datasets is shown in Table 1.

Existing CoVR models, trained on appearance-centric data, usually obtain video representations by simply averaging frame embeddings, thereby discarding temporal structure. Fine-grained retrieval demands video embeddings that preserve these dynamics. To this end we introduce a strong baseline, *TF-CoVR-Base*. Unlike recent video-language systems that depend on large-scale descriptive caption rewriting with LLMs, TF-CoVR-Base follows a concise two-stage pipeline. *Stage 1* pre-trains a video encoder on fine-grained action classification, producing temporally discriminative embeddings. *Stage 2* forms a composed query by concatenating the query-video embedding with the text-modification embedding and aligns it with candidate video embeddings via contrastive learning.

We benchmark TF-CoVR with image-based CoIR baselines, video-based CoVR systems, and general multimodal embedding (GME) models such as E5-V, evaluating every method in both zero-shot and

Table 1: Comparison of existing datasets for composed image and video retrieval, highlighting the unique features of TF-CoVR. Datasets are categorized by modality (Type), where 📷 indicates image-based and 🎥 indicates video-based triplets.

| Dataset | Type | #Triplets | Train | Eval | Multi-GT | Eval Metrics | #Sub-actions |
|---------|------|-----------|-------|------|----------|--------------|--------------|
| CIRR [24] | 📷 | 36K | ✓ | ✓ | ✗ | Recall@K | ✗ |
| FashionIQ [44] | 📷 | 30K | ✓ | ✓ | ✗ | Recall@K | ✗ |
| CC-CoIR [41] | 📷 | 3.3M | ✓ | ✗ | ✗ | Recall@K | ✗ |
| MTCIR [12] | 📷 | 3.4M | ✓ | ✗ | ✗ | Recall@K | ✗ |
| WebVid-CoVR [41] | 🎥 | 1.6M | ✓ | ✓ | ✗ | Recall@K | ✗ |
| EgoCVR [9] | 🎥 | 2K | ✗ | ✓ | ✗ | Recall@K | ✗ |
| FineCVR [50] | 🎥 | 1M | ✓ | ✓ | ✗ | Recall@K | ✗ |
| CIRCO [3] | 📷 | 800 | ✗ | ✓ | ✓ | mAP@K | ✗ |
| TF-CoVR (Ours) | 🎥 | 180K | ✓ | ✓ | ✓ | mAP@K | 306 |

fine-tuned regimes. TF-CoVR-Base attains 7.51 mAP@50 in the zero-shot setting, surpassing the best GME model (E5-V, 5.22) and all specialized CoVR methods. Fine-tuning further lifts performance to 27.22 mAP@50, a sizeable gain over the previous state-of-the-art $\text{BLIP}_{\text{CoVR-ECDE}}$ (19.83). These results underscore the need for temporal granularity and motion-aware supervision in CoVR, factors often missing in current benchmarks. TF-CoVR provides the scale to support this and exposes the limitations of appearance-based models.

To summarize, our main contributions are as follows:

- We introduce *TF-CoVR*, a large-scale benchmark for composed video retrieval centered on sports actions. The dataset comprises 180K training triplets and a test set where each query is associated with an average of 3.9 valid targets, enabling more realistic and challenging evaluation.
- We propose *TF-CoVR-Base*, a simple yet strong baseline that captures temporally fine-grained visual cues without relying on descriptive, LLM-generated captions.
- We provide the first comprehensive study of image, video, and GME models on temporally fine-grained composed retrieval under both zero-shot and fine-tuning protocols, where TF-CoVR-Base yields consistent gains across settings.

## 2 Related Work

**Video Understanding and Fast-Paced Datasets:** Video understanding [25] often involves classifying videos into predefined action categories [11, 16, 39]. These tasks are broadly categorized as coarse- or fine-grained. Coarse-grained datasets like Charades [34] and Breakfast [17] capture long, structured activities, but lack the temporal resolution and action granularity needed for composed retrieval. In contrast, fine-grained datasets like FineGym [32] and FineDiving [46] provide temporally segmented labels for sports actions. They cover high-motion actions where subtle differences (e.g., twists or apparatus) lead to semantic variation, making them suitable for retrieval tasks with fine-grained temporal changes. Yet these datasets remain unexplored in the CoVR setting, leaving a gap in leveraging temporally rich datasets. *TF-CoVR* bridges this gap by introducing a benchmark that explicitly targets temporally grounded retrieval in fast-paced, fine-grained video settings.

**Composed Image Retrieval:** CoIR retrieves a target image using a query image and a modification text describing the desired change. CoIR models are trained on large-scale triplets of query image, modification text, and target image [42, 7, 18], which have proven useful for generalizing across open-domain retrieval. CIRR [24] provides 36K curated triplets with human-written modification texts for CoIR, but it suffers from false negatives and query mismatches. CIRCO [2] improves on this by using COCO [20] and supporting multiple valid targets per query. More recently, CoLLM [12] released MTCIR, a 3.4M triplet dataset with natural captions and diverse visual scenes, addressing the lack of large-scale, non-synthetic data. Despite recent progress, existing CoIR datasets remain inherently image-centric and lack temporal depth, which restricts their applicability to video retrieval tasks requiring fine-grained temporal alignment.

**Composed Video Retrieval:** WebVid-CoVR [41] first introduced CoVR as a video extension of CoIR, using query-modification-target triplets sampled from open-domain videos. However, its lack of temporal grounding limits WebVid-CoVR's effectiveness in retrieving videos based on fine-grained

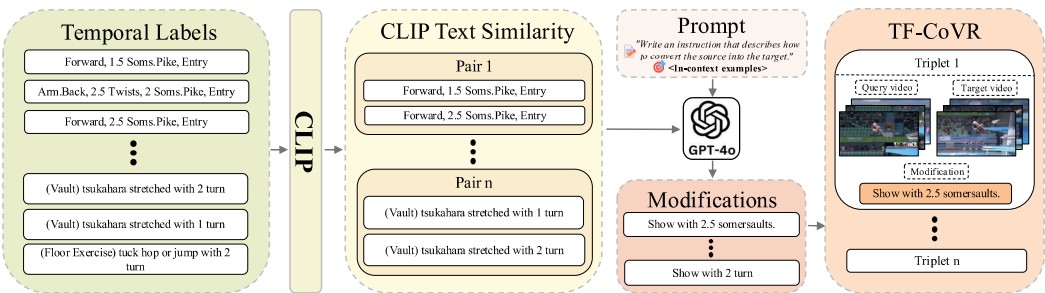

Figure 2: Overview of our automatic triplet generation pipeline for TF-CoVR. We start with temporally labeled clips from FineGym and FineDiving datasets. Using CLIP-based text embeddings, we compute similarity between temporal labels and form pairs with high semantic similarity. These label pairs are passed to GPT-4o along with in-context examples to generate natural language modifications describing the temporal differences between them. Each generated triplet consists of a query video, a target video, and a modification text capturing fine-grained temporal action changes.

action changes. EgoCVR [9] addressed this by constructing triplets within the same egocentric video to capture temporal cues. FineCVR [50] advanced CoVR by constructing a fine-grained retrieval benchmark using existing video understanding datasets such as ActivityNet [4], ActionGenome [14], HVU [6], and MSR-VTT [45]. Additionally, it introduced a consistency attribute in the modification text to guide retrieval more effectively. While an important step, the source datasets are slow-paced and coarse-grained, limiting their ability to capture subtle action transitions. Despite progress, CoVR benchmarks remain limited, relying mostly on slow-paced or object-centric content and offer only a single target per query, limiting real-world evaluation where multiple valid matches may exist.

**Multimodal Embedding Models for Composed Retrieval:** Recent advances in MLLMs such as GPT-4o [10], LLaVa [22, 21], and QwenVL [43] have significantly accelerated progress in joint visual-language understanding and reasoning tasks [31, 5, 35, 30, 36]. VISTA [53] and MARVEL [54] extend image-text retrieval by pairing pre-trained text encoders with enhanced vision encoders to better capture joint semantics. E5-V [15] and MM-Embed [19] further improve retrieval by using relevance supervision and hard negative mining to mitigate modality collapse. Zhang et al. recently introduced GME [51], a retrieval model that demonstrates strong performance on CoIR, particularly in open-domain image-text query settings. However, GME and similar MLLM-based retrievers remain untested in CoVR, especially in fast-paced scenarios requiring fine-grained temporal alignment.

## 3 TF-CoVR: Dataset Generation

**FineGym and FineDiving for Composed Video Retrieval:** Composed video retrieval (CoVR) operates on triplets $(V_q, T_m, V_t)$, where $V_q$, $T_m$, and $V_t$ denote the query video, modification text, and target video, respectively. Prior works [41, 9] construct such triplets by comparing captions and selecting pairs that differ by a small textual change, often a single word. This approach, however, relies on the availability of captions, which limits its applicability to datasets without narration. To overcome this, we use FineGym [32] and FineDiving [46], which contain temporally annotated segments but no captions. Instead of captions, we utilize the datasets' fine-grained temporal labels, which describe precise sub-actions. FineGym provides 288 labels over 32,697 clips (avg. 1.7s), from 167 long videos, and FineDiving includes 52 labels across 3,000 clips.

To identify meaningful video pairs, we compute CLIP-based similarity scores between all temporal labels and select those with high semantic similarity [27]. These pairs are then manually verified and categorized into two types: (1) temporal changes, where the sub-action differs within the same event (e.g., *(Vault) round-off, flic-flac with 0.5 turn on, stretched salto forward with 0.5 turn off* vs. *...with 2 turn off*), and (2) event changes, where the same sub-action occurs in different apparatus contexts (e.g., *(Floor Exercise) switch leap with 1 turn* vs. *(Balance Beam) switch leap with 1 turn*). These examples show that even visually similar actions can have different semantic meanings depending on temporal or contextual cues. We apply this strategy to both FineGym and FineDiving to generate rich, fine-grained video triplets. (See Figure 1 for illustrations.)

**Modification Instruction and Triplet Generation:** To generate modification texts for TF-CoVR, we start with the fine-grained temporal labels associated with gymnastics and diving segments, such

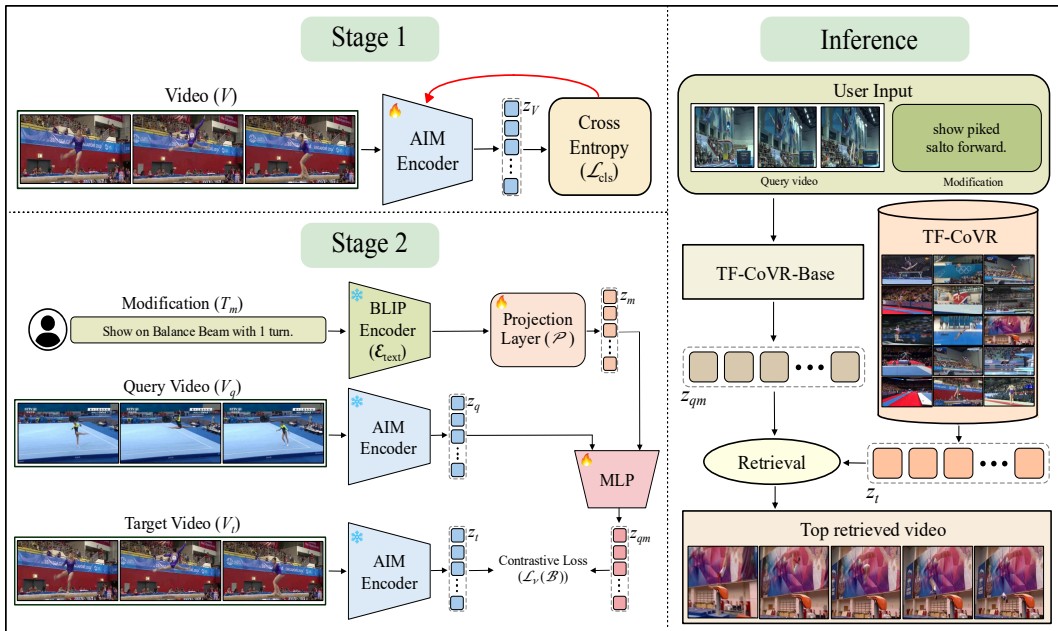

Figure 3: Overview of TF-CoVR-Base framework. Stage 1 learns temporal video representations via supervised classification using the AIM encoder. In Stage 2, the pretrained AIM and BLIP encoders are frozen, and a projection layer and MLP are trained to align the query-modification pair with the target video using contrastive loss. During inference, the model retrieves relevant videos from TF-CoVR based on a user-provided query video and textual modification.

as *Forward, 1.5 Soms.Pike, Entry* or *(Vault) tsukahara stretched with 2 turn*. Using CLIP, we compute pairwise similarity scores between all labels and select those that differ in small but meaningful aspects, representing source and target actions connected by a semantic modification.

Each selected label pair is passed to GPT-4o [10] along with a prompt and 15 in-context examples capturing typical sub-action and event-level changes [40]. GPT-4o generates concise natural language instructions that describe how to transform the source into the target, e.g., *Show with 2.5 somersaults* or *Show on Balance Beam*. Unlike prior work such as FineCVR [50], which emphasizes visual consistency, our modifications focus exclusively on temporal changes, making them better suited for real-world use cases like highlight generation where visual similarity is not required.

To form triplets, we split the original long-form videos into training and testing sets to avoid overlap. From these, sub-action clips are extracted and paired with the corresponding modification text. Although individual clips may be reused, each resulting triplet, comprising a query video, a modification text, and a target video, is unique. This process is repeated exhaustively across all labeled segments. Figure 2 illustrates the full pipeline, from label pairing to triplet generation.

**TF-CoVR Statistics:** TF-CoVR contains 180K training triplets and 473 testing queries, each associated with multiple ground-truth target videos (Table 1). The test set specifically addresses the challenge of evaluating multiple valid retrievals, a limitation in existing CoVR benchmarks. The dataset spans 306 fine-grained sports actions: 259 from FineGym [32] and 47 from FineDiving [46]. Clip durations range from 0.03s to 29.00s, with an average of 1.90s.

Modification texts vary from 2 to 19 words (e.g., *"show off"* to *"Change direction to Reverse, reduce to two and a half twists, and show with one and a half somersaults"*), with an average length of 6.11 words. Each test query has an average of 3.94 valid targets, supporting realistic and challenging evaluation under a multi-ground-truth setting. This makes TF-CoVR suited for applications like highlight generation in sports broadcasting, where retrieving diverse sub-action variations is essential.

## 4 TF-CoVR-Base: Structured Temporal Learning for CoVR

**Method Overview:** In the composed video retrieval (CoVR) task, the goal is to retrieve a target video $V_t$ given a query video $V_q$ and a textual modification $T_m$ that describes the intended transformation.

Table 2: Benchmarking results on TF-CoVR using mAP@K for $K \in \{5, 10, 25, 50\}$. We evaluate two groups of models: (1) *Existing CoVR methods trained on WebVid-CoVR and not fine-tuned on TF-CoVR*, and (2) *General Multimodal Embeddings*, tested in a zero-shot setting. Each model is evaluated on query-target pairs consisting of the specified number of sampled frames. "CA" denotes the use of cross-attention fusion.

| Modalities | | Model | Fusion | #Query | #Target | mAP@K ($\uparrow$) | | | |
|---|---|---|---|---|---|---|---|---|---|
| Video | Text | | | Frames | Frames | 5 | 10 | 25 | 50 |
| *General Multimodal Embeddings (TF-CoVR)* | | | | | | | | | |
| ✓ | ✓ | GME-Qwen2-VL-2B [51] | MLLM | 1 | 15 | 2.28 | 2.64 | 3.29 | 3.81 |
| ✓ | ✓ | MM-Embed [19] | MLLM | 1 | 15 | 2.39 | 2.81 | 3.61 | 4.14 |
| ✓ | ✓ | E5-V [15] | Avg | 1 | 15 | 3.14 | 3.78 | 4.65 | 5.22 |
| *Not fine-tuned on TF-CoVR* | | | | | | | | | |
| ✗ | ✓ | BLIP2 | - | - | 15 | 1.34 | 1.79 | 2.20 | 2.50 |
| ✓ | ✗ | BLIP2 | - | 1 | 15 | 1.74 | 2.20 | 3.06 | 3.62 |
| ✓ | ✓ | BLIP-CoVR [41] | CA | 1 | 15 | 2.33 | 2.99 | 3.90 | 4.50 |
| ✓ | ✓ | BLIP$_{\text{CoVR-ECDE}}$ [37] | CA | 1 | 15 | 0.78 | 0.88 | 1.16 | 1.37 |
| ✗ | ✓ | TF-CVR [9] | - | - | 15 | 0.56 | 0.76 | 0.99 | 1.24 |
| ✓ | ✓ | LanguageBind [55] | Avg | 8 | 8 | 3.43 | 4.37 | 5.26 | 5.92 |

This requires learning a cross-modal relationship between visual and textual inputs that captures how the target differs from the query. While prior methods have shown promise on general video datasets, TF-CoVR becomes significantly more challenging in fine-grained, fast-paced domains such as gymnastics and diving, where subtle temporal action differences are critical. Existing approaches often overlook these dynamics, motivating the need for a more temporally grounded framework.

**Two-Stage CoVR Approach:** We propose a two-stage training framework, TF-CoVR-Base, for composed video retrieval in fine-grained, fast-paced domains such as gymnastics and diving. TF-CoVR-Base is designed to explicitly capture the temporal structure in videos and align it with textual modifications for accurate retrieval. Unlike prior approaches that rely on average-pooled frame features from image-level encoders, TF-CoVR-Base decouples temporal representation learning from the retrieval task. It first learns temporally rich video embeddings through supervised action classification, and then uses these embeddings in a contrastive retrieval setup. We describe each stage of the framework below.

**Stage One: Temporal Pretraining via Video Classification:** In the first stage, we aim to learn temporally rich video representations from TF-CoVR. To this end, we employ the AIM encoder [48], which is specifically designed to capture temporal dependencies by integrating temporal adapters into a CLIP-based backbone.

We pretrain the AIM encoder on a supervised video classification task using all videos from the triplets in the training set. Let $V = \{f_1, f_2, \ldots, f_f\}$ denote a video clip with $f$ frames. The AIM encoder processes each frame and produces a sequence-level embedding:

$$z_V = \text{AIM}(V).$$

The classification logits $z_V$ are passed through a softmax function to produce a probability distribution over classes:

$$\hat{p}_V^{(i)} = \text{Softmax}(z_V^{(i)}).$$

Each video $V$ is annotated with a ground-truth label $y_V$, and the model is optimized using the standard cross-entropy loss:

$$\mathcal{L}_{\text{cls}} = -\sum_{i=1}^{C} y_V^{(i)} \log \hat{p}_V^{(i)}.$$

where $C = 306$ is the total number of fine-grained action classes in the TF-CoVR dataset.

**Stage Two: Contrastive Training for Retrieval:** In the second stage of TF-CoVR-Base, we train a contrastive model to align the composed query representations with the target video representations.

As illustrated in Figure 3, each training sample is structured as a triplet $(V_q, T_m, V_t)$, where $V_q$ is the **query video** consisting of $N$ frames, $T_m$ is the **modification text** with $L$ tokens, and $V_t$ is the **target video** comprising $M$ frames.

We use our pretrained and frozen AIM encoder from stage 1 to extract temporally rich embeddings for the query and target videos:

$$z_q = \text{AIM}(V_q), \quad z_t = \text{AIM}(V_t).$$

The modification text $T_m$ is encoded using the BLIP2 text encoder $\mathcal{E}_{\text{text}}$, followed by a learnable projection layer $\mathcal{P}$ that maps the text embedding into a shared embedding space. This step ensures the textual features are adapted and aligned with the video modality for the CoVR task:

$$z_m = \mathcal{P}(\mathcal{E}_{\text{text}}(T_m)).$$

We then fuse the query video embedding $z_q$ and the projected text embedding $z_m$ using a multi-layer perceptron (MLP), producing the composed query representations:

$$z_{qm} = \text{MLP}(z_q, z_m).$$

To compare the composed query embeddings with the target video embeddings, both $z_{qm}$ and $z_t$ are projected into a shared embedding space and normalized to unit vectors. Their relationship is then measured using cosine, computed as:

$$S_{i,j} = \frac{z_{qm}^{(i)} \cdot z_t^{(j)}}{\|z_{qm}^{(i)}\| \, \|z_t^{(j)}\|}.$$

To ensure numerical stability and regulate the scale of similarity scores, cosine similarity is adjusted using a temperature parameter:

$$\text{sim}(z_{qm}^{(i)}, z_t^{(j)}) = \frac{S_{i,j}}{\tau}.$$

where $\tau \in \mathbb{R}_{>0}$ is the temperature parameter. We then define a scaled similarity matrix $\tilde{S}$ using a concentration parameter $\beta \geq 0$:

$$\tilde{S}_{i,j} = \beta \cdot S_{i,j}.$$

The weight assigned to each negative sample in the loss is computed using a softmax-like reweighting scheme, with diagonal entries (positive pairs) scaled by a hyperparameter $\alpha \in (0, 1]$:

$$w_{i,j}^{i \to t} = \begin{cases} \alpha, & \text{if } i = j \\ \dfrac{(n-1) \cdot \exp(\tilde{S}_{i,j})}{\sum\limits_{k \neq i} \exp(\tilde{S}_{i,k})}, & \text{otherwise} \end{cases} \qquad w_{j,i}^{t \to i} = \begin{cases} \alpha, & \text{if } j = i \\ \dfrac{(n-1) \cdot \exp(\tilde{S}_{j,i})}{\sum\limits_{k \neq i} \exp(\tilde{S}_{k,i})}, & \text{otherwise} \end{cases}$$

Finally, the HN-NCE loss [29] is defined as followed, which emphasizes hard negatives by assigning greater weights to semantically similar but incorrect targets. Given a batch $\mathcal{B}$ of triplets $(q_i, m_i, t_i)$, the loss is defined as:

$$\mathcal{L}_v(\mathcal{B}) = \frac{1}{n} \sum_{i=1}^{n} \left[ \log \left( \sum_{j=1}^{n} \exp(S_{i,j}) \cdot w_{i,j}^{i \to t} \right) + \log \left( \sum_{j=1}^{n} \exp(S_{j,i}) \cdot w_{j,i}^{t \to i} \right) - 2 S_{i,i} \right].$$

Here, $S_{i,j}$ is the cosine similarity between the composed query $z_{qm}^{(i)}$ and the target video $z_t^{(j)}$, $\alpha$ is a scalar constant (set to 1), $\tau$ is a temperature hyperparameter (set to 0.07). In our experiments, we set $\alpha = 1$ and $\beta = 0$, reducing the formulation to the standard InfoNCE [28] loss.

Table 3: Evaluation of models fine-tuned on TF-CoVR using mAP@K for $K \in \{5, 10, 25, 50\}$. We report the performance of various fusion strategies and model architectures trained on TF-CoVR. Fusion methods include MLP and cross-attention (CA). Each model is evaluated using a fixed number of sampled frames from both query and target videos. Fine-tuning on TF-CoVR leads to significant improvements across all models. The results for TF-CoVR-Base (Stage-2 only) reflect the model's performance without Stage-1 temporal pretraining.

| Modalities | | Model | Fusion | #Query | #Target | mAP@K ($\uparrow$) | | | |
|---|---|---|---|---|---|---|---|---|---|
| Video | Text | | | Frames | Frames | 5 | 10 | 25 | 50 |
| | | | *Fine-tuned on TF-CoVR* | | | | | | |
| ✗ | ✓ | BLIP2 | - | - | 15 | 10.69 | 13.02 | 15.35 | 16.41 |
| ✓ | ✗ | BLIP2 | - | 1 | 15 | 4.86 | 6.49 | 8.92 | 10.06 |
| ✓ | ✓ | CLIP | MLP | 1 | 15 | 7.01 | 8.35 | 10.22 | 11.38 |
| ✓ | ✓ | BLIP2 | MLP | 1 | 15 | 10.86 | 13.20 | 15.38 | 16.31 |
| ✓ | ✓ | CLIP | MLP | 15 | 15 | 6.40 | 7.46 | 9.21 | 10.40 |
| ✓ | ✓ | BLIP2 | MLP | 15 | 15 | 11.64 | 14.81 | 16.74 | 17.55 |
| ✓ | ✓ | BLIP-CoVR | CA [41] | 1 | 15 | 11.07 | 13.94 | 16.07 | 16.88 |
| ✓ | ✓ | BLIP$_{\text{CoVR-ECDE}}$ | CA [37] | 1 | 15 | 13.03 | 15.90 | 18.62 | 19.83 |
| ✓ | ✓ | TF-CoVR-Base (Stage-2 only) | MLP | 8 | 8 | 15.08 | 18.70 | 21.78 | 22.61 |
| ✓ | ✓ | TF-CoVR-Base (Ours) | MLP | 12 | 12 | **21.85** | **24.23** | **26.47** | **27.22** |

## 5  Discussion

**Evaluation Metric:** To effectively evaluate retrieval performance in the presence of multiple ground-truth target videos, we adopt the mean Average Precision at $K$ (mAP@K) metric, as proposed in CIRCO [3]. The mAP@K metric measures whether the correct target videos are retrieved and considers the ranks at which they appear in the retrieval results.

Here, $K$ denotes the number of top-ranked results considered for evalua-

Table 4: Performance of GME models on existing CoIR benchmarks. We report mAP@5 and Recall@10 on FashionIQ, CIRR, and CIRCO using official evaluation protocols. Values are directly taken from the original papers.

| Model | Metric | FQ | CIRR | CIRCO |
|---|---|---|---|---|
| E5-V [15] | Recall@10 | 3.73 | 13.19 | - |
| GME-2B [51] | Recall@10 | 26.34 | 47.70 | - |
| MM-Embed [19] | Recall@10 | 25.7 | 50.0 | - |
| E5-V [15] | mAP@5 | - | - | 19.1 |
| MM-Embed [19] | mAP@5 | - | - | 32.3 |

tion. For example, mAP@5 measures precision based on the top 5 retrieved videos, capturing how well the model retrieves relevant targets early in the ranked list. A higher $K$ allows evaluation of broader retrieval quality, while a lower $K$ emphasizes top-ranking precision.

**Specialized vs. Generalized Multimodal Models for CoVR:** We compare specialized models trained specifically for composed video retrieval, such as those trained on WebVid-CoVR [41], with Generalized Multimodal Embedding (GME) models that have not seen CoVR data. Among the specialized baselines, we include two image-based encoders (CLIP and BLIP) and one video-based encoder (LanguageBind) to cover different modality types and fusion mechanisms. As shown in Table 2, our evaluation reveals that GME models consistently outperform most specialized CoVR methods in the zero-shot setting. For example, E5-V [15] achieves 5.22 mAP@50, outperforming BLIP-CoVR (4.50) and BLIP$_{\text{CoVR-ECDE}}$ (1.37), and closely matching LanguageBind (5.92). Other GME variants like MM-Embed and GME-Qwen2-VL-2B also show promising results. In contrast, TF-CVR [9] performs worst among all tested models, with only 1.24 mAP@50, underscoring its limitations in handling fine-grained action variations.

This performance gap is partly due to TF-CVR's reliance on a captioning model to describe the query video. We replaced the original Lavila [52] with Video-XL [33], which provides better captions for structured sports content. However, even Video-XL fails to capture subtle temporal cues like twist counts or somersaults, critical for accurate retrieval, causing TF-CVR to struggle with temporally precise matches. In contrast, GME models benefit from large-scale multimodal training involving text, images, and combinations thereof, allowing them to generalize well to CoVR without task-

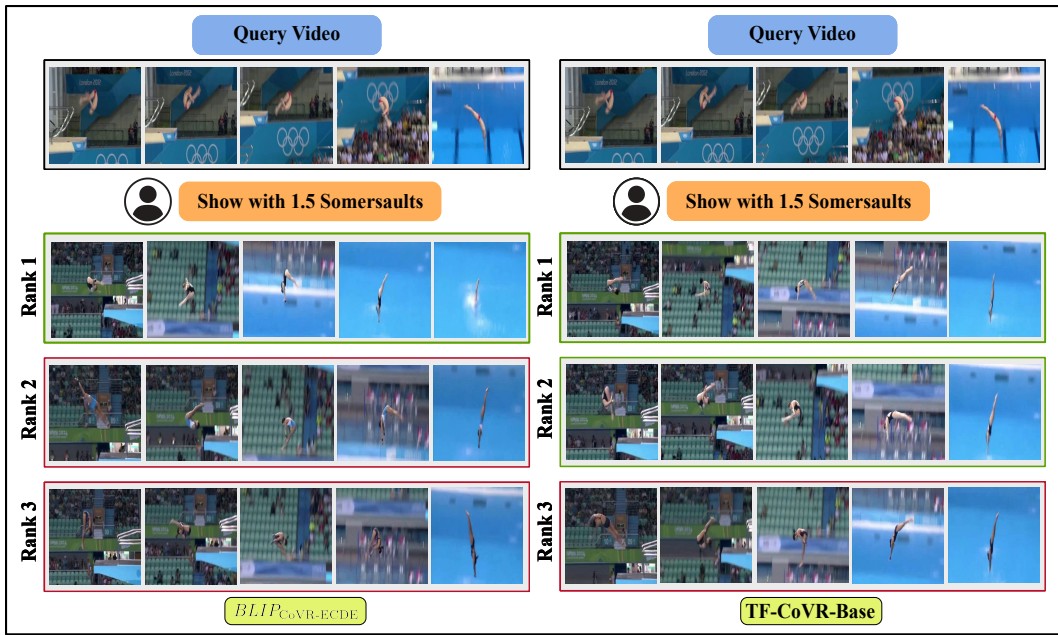

Figure 4: Qualitative results for the composed video retrieval task using our two-stage TF-CoVR-Base model. Each column showcases a query video (top), the corresponding modification instruction (middle), and the top-3 retrieved target videos (ranks 1–3) based on model predictions. TF-CoVR-Base effectively captures subtle temporal variations and retrieves the correct target video at higher ranks. In contrast, the baseline method BLIP_{CoVR-ECDE} often fails to identify the correct action class or resolve fine-grained temporal differences, as indicated by the errors highlighted in red.

specific fine-tuning. We expect their performance to improve further with fine-tuning on TF-CoVR, though we leave this exploration to future work. See supplementary material for a comparison of Lavila-generated captions.

**Evaluating TF-CoVR-Base Against Existing Methods:** We compare our proposed two-stage TF-CoVR-Base framework with all existing CoVR baselines in Table 3. Our full model achieves 27.22 mAP@50, significantly outperforming the strongest prior method, BLIP_{CoVR-ECDE} (19.83). Even our Stage-2-only variant (trained without temporal pretraining) outperforms all existing methods with 22.61 mAP@50, highlighting the strength of our contrastive fusion strategy. Unlike BLIP_{CoVR-ECDE}, our model does not rely on detailed textual descriptions of the query video and instead learns temporal structure directly from the visual input. This makes it especially effective in structured, fast-paced sports videos, where subtle action distinctions, such as change in twist count or apparatus, are visually grounded. Across all K values, TF-CoVR-Base shows consistent improvements of 4-6 mAP points.

**Impact of Hard-Negative Weighting on TF-CoVR:** We further investigate the impact of hard-negative (HN) weighting in the HN-NCE loss function. Specifically, we compare different weighting values, including the baseline setting of 0, which reduces the loss to the standard InfoNCE [28] formulation. Our re-

Table 5: Performance comparison between the HN-NCE and InfoNCE loss by varying the HN-weighting.

| HN-Weighting | mAP@5 | mAP@10 | mAP@25 | mAP@50 |
|---|---|---|---|---|
| 0.7 | 20.40 | 22.46 | 24.63 | 25.37 |
| 0.5 | 21.02 | 22.89 | 25.21 | 25.91 |
| 0.3 | 20.86 | 23.35 | 25.44 | 26.16 |
| 0.0 | 21.85 | 24.23 | 26.47 | 27.22 |

sults show that InfoNCE (HN-weighting = 0) consistently outperforms the HN-NCE variants with positive weighting values. While HN-NCE is designed to emphasize hard negatives by assigning them higher weights, this approach can introduce optimization noise, particularly in fine-grained settings where many negative samples are visually similar to the positives. In such scenarios, treating all negatives equally, as in InfoNCE, appears to provide more stable training and better discrimination based on subtle visual cues. As shown in Table 5, reducing the HN-weighting from 0.7 to 0.0 results in a performance gain from 25.37 to 27.22 mAP, an increase of over 1.8 mAP points.

**Qualitative Analysis:** Figure 4 illustrates the effectiveness of our method using qualitative examples. The retrieved target videos accurately reflect the action modifications described in the input text. Correctly retrieved clips are outlined in green, and incorrect ones in red. Interestingly, even incorrect predictions are often semantically close to the intended target, revealing the fine-grained difficulty of TF-CoVR. For example, in the third column of Figure 4, the query video includes a turning motion, while the modification requests a *"no turn"* variation. Our method correctly retrieves *"no turn"* actions at top ranks, but at rank 3, retrieves a "split jump" video, visually similar but semantically different. We highlight this with a red overlay to emphasize the subtle distinction in motion, showing the value of TF-CoVR for evaluating fine-grained temporal reasoning.

**Domain-Specific Pretraining for Temporal Reasoning:** Although TF-CoVR-Base is designed to be *domain agnostic*, its current training leverages domain-specific datasets to better capture the fine-grained and structured nature of different activity domains, such as surgery or daily tasks. Domain-specific pretraining proves beneficial for learning distinct temporal patterns and visual cues inherent to each domain. For example, in a surgical setting, a query video may depict a sequence such as *"insert needle at a 30-degree angle, advance 2 cm, then begin the suture loop with the right hand,"* while the corresponding target video modifies this to *"insert needle at a 45-degree angle, advance 3 cm, then begin the suture loop with the right hand."* The modification text, *"change needle insertion angle to 45 degrees and advance by 3 cm instead of 2 cm,"* captures subtle changes in motion angle and depth. Accurately modeling such fine-grained temporal variations necessitates temporally discriminative features, which are challenging to learn without domain-specific pretraining. This positions TF-CoVR-Base to provide a strong foundation for exploring more generalizable temporal reasoning methods across diverse and less-structured video domains.

# 6   Limitations and Conclusion

**Limitations.** TF-CoVR offers a new perspective on composed video retrieval by focusing on retrieving videos that reflect subtle action changes, guided by a modification text. While it adds valuable depth to the field, the dataset has some limitations. One limitation is that it requires expert effort to temporally annotate videos such as from FineGym and FineDiving, which is currently lacking in the video-understanding community, and such annotation is expensive to scale up. This reflects the trade-off between expert-driven annotations and scalability. Regarding the TF-CoVR-Base, it is currently two-stage, which may not provide a fully end-to-end solution; a better approach could be a single-stage model that simultaneously learns temporally rich video representations and aligns them with the modification text.

**Conclusion.** In this work, we introduced TF-CoVR, a large-scale dataset comprising 180K unique triplets centered on fine-grained sports actions, spanning 306 diverse sub-actions from gymnastics and diving videos. TF-CoVR brings a new dimension to the CoVR task by emphasizing subtle temporal action changes in fast-paced, structured video domains. Unlike existing CoVR datasets, it supports multiple ground-truth target videos per query, addressing a critical limitation in current benchmarks and enabling more realistic and flexible evaluation. In addition, we propose a two-stage training framework that explicitly models temporal dynamics through supervised pre-training. Our method significantly outperforms existing approaches on TF-CoVR. Furthermore, we conducted a comprehensive benchmarking of both existing CoVR methods and General Multimodal Embedding (GME) models, marking the first systematic evaluation of GME performance in the CoVR setting. We envision TF-CoVR serving as a valuable resource for real-world applications such as sports highlight generation, where retrieving nuanced sub-action variations is essential for generating engaging and contextually rich video content.

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
