# Supplementary Material

## A TF-CoVR Statistics and Modification Lexicon

**TF-CoVR Statistics** We present detailed statistics on the distribution of video counts per label in *TF-CoVR*, which comprises a diverse set of 306 annotated sub-actions. Figures A1 and A2 show the label-wise video distribution for the *FineGym* [3] and *FineDiving* [6] subsets of *TF-CoVR*, respectively. Both distributions are plotted on a logarithmic scale to emphasize the long-tailed nature of label frequencies. In *FineGym*, many labels have several hundred to over a thousand associated videos, with a gradual decline across the distribution. This results in broad coverage of fine-grained sub-actions. By contrast, *FineDiving* exhibits a steeper drop in video count per label, primarily due to its smaller dataset size. Nevertheless, a substantial number of labels still contain more than 30 samples, preserving enough diversity to support *temporal fine-grained composed video retrieval. TF-CoVR* thus serves as a strong benchmark for learning and evaluating fine-grained temporal reasoning in the composed video retrieval task.

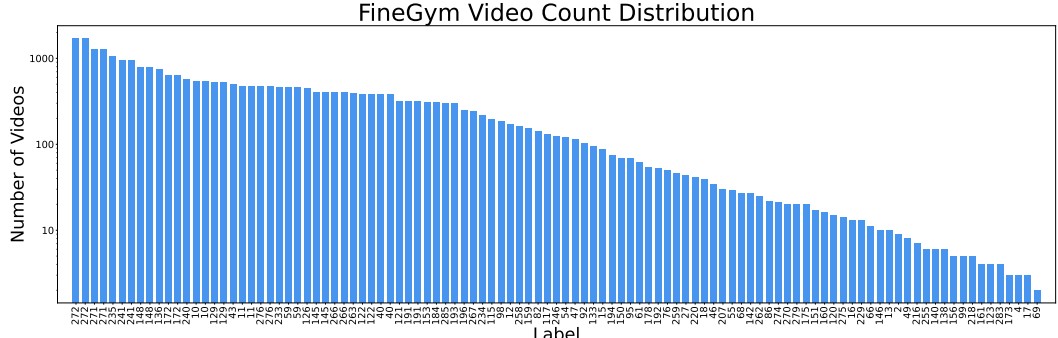

Figure A1: Label-wise video count distribution in the *FineGym* subset of *TF-CoVR*. A logarithmic scale is used on the y-axis to highlight the steep drop in video counts per label due to the smaller dataset size. Note that only a subset of all labels is shown for clarity.

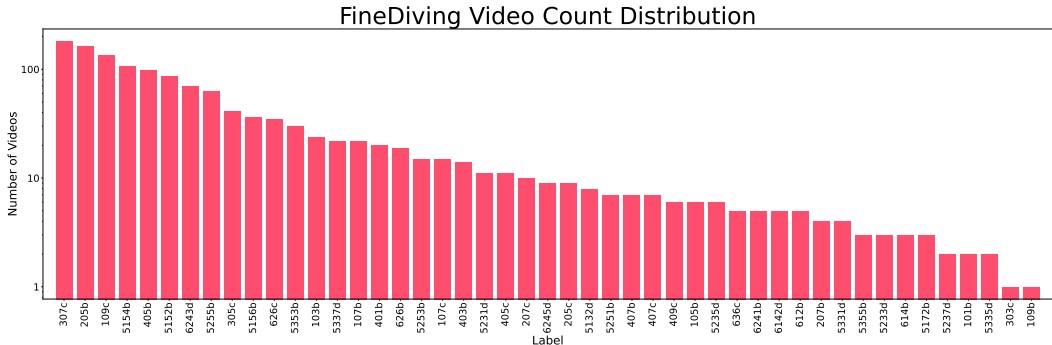

Figure A2: Label-wise video count distribution in the *FineDiving* subset of *TF-CoVR*. The y-axis is plotted on a logarithmic scale to highlight the steep drop in video counts per label due to the smaller dataset size, while still preserving label diversity.

**Modification Lexicon** Figure A3 presents a word cloud visualization of the most frequently occurring terms in the modification texts of *TF-CoVR*. Prominent terms such as *twist*, *turn*, *salto*, *backward*, *tucked*, *stretched*, and *piked* highlight the fine-grained, motion-centric nature of these

modifications. These terms encapsulate key action semantics related to orientation, body posture, and movement complexity, covering aspects such as the number of twists or turns, in-air body position, and directional shifts like forward or backward. The presence of apparatus-specific terms such as *Beam*, *Floor*, and *Exercise* further underscores the diversity of event contexts represented in the dataset. This rich and structured lexicon enables *TF-CoVR* to support nuanced temporal modifications, distinguishing it from existing datasets that often rely on coarser or less temporally dynamic instructions.

In this appendix, we provide more details, experimental results, qualitative visualization of our new *TF-CoVR* dataset and our two-stage *TF-CoVR-Base* method.

## B    TF-CoVR: Modification Text Generation

To support *TF-CoVR* modification generation, we craft domain-adapted prompting strategies for GPT-4o [2], addressing the unique structure of gymnastics and diving videos. Given the structural differences between *FineGym* [3] and *FineDiving* [6], we developed separate prompts for each domain. *Fin-*

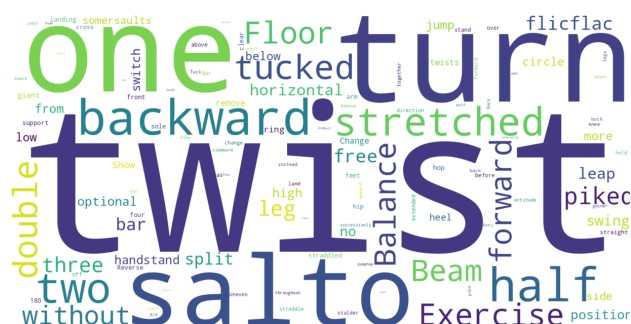

Figure A3: Word cloud visualization of the most frequent action-related terms in *TF-CoVR* modification texts. Larger words indicate higher frequency and reflect the dataset's fine-grained, motion-centric nature, with terms like *twist*, *turn*, *salto*, and apparatus names such as *Beam* and *Floor* highlighting contextual diversity across domains.

*eGym*, with its substantially larger set of annotated sub-actions, was provided with 20 in-context examples to better capture the diversity and complexity of its routines. In contrast, we used 5 in-context examples for *FineDiving*, reflecting its smaller label set and more compact structure.

**Prompt and In-Context Examples**    To support accurate modification generation for *TF-CoVR*, we designed prompt templates and in-context examples that align with the linguistic and structural characteristics of the gymnastics and diving domains.

---

**Modification Generation Prompt for FineDiving**

You are an expert in designing tasks that require understanding the transformation between two description, specifically for video descriptions. Your goal is to ensure that the instructions you provide are concise, accurate, and focused on the necessary modifications between the source and target description.

**Instructions:**

1. Analyze the given source and target description.

2. Identify the changes between the source and target description.

3. Write an instruction that describes only the transformation required to achieve the target description from the source.

4. Ensure the instruction is as short as possible, focusing on actions. Mention objects only when absolutely necessary.

5. Do not describe objects or actions common to both descriptions. Use pronouns when appropriate.

6. Your response should focus only on the transformation, without extraneous details or repetitions.

**Remember:**

- Keep the instruction concise and focus only on the transformation required.

- Avoid redundant details or describing elements unchanged between source and target descriptions.

**In-Context Examples:**

> **Source Description:** Inward, 3.5 Soms.Tuck, Entry
> **Target Description:** Inward, 4.5 Soms.Tuck, Entry
> **Modification text:** Show with 4.5 somersaults Tuck.

> **Source Description:** Inward, 3.5 Soms.Tuck, Entry
> **Target Description:** Inward, 2.5 Soms.Tuck, Entry
> **Modification text:** Show with 2.5 somersaults Tuck.

> **Source Description:** Back, 1.5 Twists, 2.5 Soms.Pike, Entry
> **Target Description:** Back, 2.5 Twists, 1.5 Soms.Pike, Entry
> **Modification text:** Show with 2.5 twists and 1.5 somersaults.

> **Source Description:** Forward, 3.5 Soms.Pike, Entry
> **Target Description:** Forward, 1.5 Soms.Pike, Entry
> **Modification text:** Show with 1.5 somersaults.

> **Source Description:** Arm.Back, 2.5 Twists, 2 Soms.Pike, Entry
> **Target Description:** Arm.Back, 1.5 Twists, 2 Soms.Pike, Entry
> **Modification text:** Show with 1.5 twists.

---

**Modification Generation Prompt for FineGym**

You are an expert in designing tasks that require understanding the transformation between two description, specifically for video descriptions. Your goal is to ensure that the instructions you provide are concise, accurate, and focused on the necessary modifications between the source and target description.

**Instructions:**

1. Analyze the given source and target description.
2. Identify the changes between the source and target description.
3. Write an instruction that describes only the transformation required to achieve the target description from the source.
4. Ensure the instruction is as short as possible, focusing on actions. Mention objects only when absolutely necessary.
5. Do not describe objects or actions common to both descriptions. Use pronouns when appropriate.
6. Your response should focus only on the transformation, without extraneous details or repetitions.

**Remember:**

- Keep the instruction concise and focus only on the transformation required.
- Avoid redundant details or describing elements unchanged between source and target descriptions.

**In-Context Examples:**

**Source Narration:** (VT) round-off, flic-flac with 0.5 turn on, stretched salto forward with 1.5 turn off.
**Target Narration:** (VT) round-off, flic-flac with 0.5 turn on, stretched salto forward with 0.5 turn off.
**Instruction:** show with 0.5 turn.

**Source Narration:** (VT) round-off, flic-flac with 0.5 turn on, stretched salto forward with 1.5 turn off.
**Target Narration:** (VT) round-off, flic-flac with 0.5 turn on, stretched salto forward with 1 turn off.
**Instruction:** show with 1 turn.

**Source Narration:** (VT) round-off, flic-flac with 0.5 turn on, stretched salto forward with 1.5 turn off.
**Target Narration:** (VT) round-off, flic-flac with 0.5 turn on, 0.5 turn to piked salto backward off.
**Instruction:** show 0.5 turn with spiked salto backward.

**Source Narration:** (VT) round-off, flic-flac with 0.5 turn on, stretched salto forward with 1.5 turn off.
**Target Narration:** (VT) round-off, flic-flac with 1 turn on, piked salto backward off.
**Instruction:** show flic-flac with 1 turn and picked salto backward.

**Source Narration:** (VT) round-off, flic-flac with 0.5 turn on, stretched salto forward with 0.5 turn off.
**Target Narration:** (VT) round-off, flic-flac with 0.5 turn on, stretched salto forward with 1.5 turn off.
**Instruction:** show with 1.5 turn.

**Source Narration:** (VT) round-off, flic-flac with 0.5 turn on, piked salto forward off.
**Target Narration:** (VT) round-off, flic-flac with 0.5 turn on, stretched salto forward with 1.5 turn off.
**Instruction:** show stretched salto forward with 1.5 turn.

**Source Narration:** (VT) round-off, flic-flac with 0.5 turn on, piked salto forward off.
**Target Narration:** (VT) round-off, flic-flac with 1 turn on, piked salto backward off.
**Instruction:** show flic-flac with 1 turn and piked salto backward.

**Source Narration:** (VT) round-off, flic-flac with 1 turn on, piked salto backward off.
**Target Narration:** (VT) round-off, flic-flac with 0.5 turn on, piked salto forward off.
**Instruction:** show flic-flac with 0.5 turn and piked salto forward.

**Source Narration:** (VT) tsukahara stretched with 2 turn.
**Target Narration:** (VT) tsukahara stretched with 1 turn.
**Instruction:** show with 1 turn.

**Source Narration:** (VT) tsukahara stretched with 2 turn.
**Target Narration:** (VT) tsukahara tucked with 1 turn.
**Instruction:** show tucked with 1 turn.

**Source Narration:** (VT) tsukahara stretched salto.
**Target Narration:** (VT) tsukahara stretched without salto.
**Instruction:** show without salto.

**Source Narration:** (FX) switch leap with 0.5 turn.
**Target Narration:** (BB) switch leap with 0.5 turn.
**Instruction:** show on BB.

**Source Narration:** (FX) switch leap with 0.5 turn.
**Target Narration:** (FX) split jump with 0.5 turn.
**Instruction:** show a split jump.

**Source Narration:** (FX) switch leap with 0.5 turn.
**Target Narration:** (FX) switch leap.
**Instruction:** show a switch leap with no turn.

**Source Narration:** (FX) switch leap with 1 turn.
**Target Narration:** (BB) split leap with 1 turn.
**Instruction:** show a split leap on BB.

**Source Narration:** (FX) stag jump.
**Target Narration:** (FX) stag ring jump.
**Instruction:** show with ring.

**Source Narration:** (FX) tuck hop or jump with 1 turn.
**Target Narration:** (FX) wolf hop or jump with 1 turn.
**Instruction:** show wolf hop.

**Source Narration:** (FX) pike jump with 1 turn.
**Target Narration:** (BB) straddle pike jump with 1 turn.
**Instruction:** show straddle pike jump on BB.

**Source Narration:** (UB) (swing forward) salto backward stretched.
**Target Narration:** (UB) (swing backward) double salto forward tucked with 0.5 turn.
**Instruction:** show (swing backward) double salto forward tucked with 0.5 turn.

**Source Narration:** (UB) (swing forward) double salto backward stretched with 1 turn.
**Target Narration:** (UB) (swing forward) salto backward stretched with 2 turn.
**Instruction:** show salto backward stretched with 2 turn.

## C  Limitations of Existing Captioning Models

We present a detailed comparison between the captions generated by existing video captioning models and the structured descriptions curated for our *TF-CoVR* dataset. As *TF-CoVR* is designed around triplets centered on fine-grained temporal actions, it is essential that captioning models capture key elements such as action type, number of turns, and the apparatus involved. Our analysis shows that current models, such as LaVila [7] and VideoXL [4], often fail to identify these fine-grained details, underscoring their limitations in handling temporally precise and action-specific scenarios.

**Caption Generation Template for VideoXL** To generate technically accurate captions for gymnastics and diving routines, we supply VideoXL with domain-specific prompts tailored to each sport. These prompts incorporate specialized vocabulary and structured syntax to align with official judging terminology. In both sports, subtle variations, such as differences in twist count, body position, or apparatus, convey distinct semantic meaning. To capture this level of granularity, we apply strict formatting constraints and exemplar-based guidance during prompting. While this structured approach helps VideoXL focus on fine-grained action details, the generated captions still exhibit inconsistencies and often fail to capture critical aspects of the routines with sufficient reliability.

---

**VideoXL Caption Generation Prompt for FineGym**

You are an expert gymnastics judge.
Your task is to provide a **strictly formatted, concise technical caption** for the gymnast's routine. Use **official gymnastics vocabulary only** (e.g., round-off, flic-flac, salto, tuck, pike, layout).
DO NOT describe emotions, strength, balance, or control.
DO NOT explain what it "shows" or "demonstrates."
DO NOT use generic verbs like "move", "flip", "spin", "pose", etc.
Include:
- Entry move (e.g., round-off)
- Main move (e.g., double back salto)
- Body position (e.g., tuck, layout, pike)
- Number of twists or somersaults (e.g., 1.5 twists, triple salto)
- Apparatus name if identifiable
Only output a **single-line caption**, no lists, no bullets, no extra sentences.
Format: [Technical move sequence with turns and position].
(Apparatus: [FX / VT / BB / UB / Unknown])
Examples:
- Round-off, flic-flac, double tuck salto with 1.5 twist. (Apparatus: FX)
- Back handspring to layout salto with full twist. (Apparatus: BB)
- Stretched salto backward with 2.5 twists. (Apparatus: VT)

---

**VideoXL Caption Generation Prompt for FineDiving**

You are an expert **diving judge**.
Your task is to provide a **strictly formatted, concise technical caption** for the diver's routine based on official diving terminology. Use terms defined by **FINA** and standard competition vocabulary.
DO NOT describe emotions, grace, beauty, or control. DO NOT narrate or explain what it "shows" or "demonstrates."
DO NOT use vague verbs like "moves", "flips", "spins", or any stylistic language.
Include:
- Takeoff direction (e.g., forward, backward, reverse, inward, armstand)
- Number of somersaults (e.g., 1.5, 2.5, 3.5)
- Number of twists (if any)
- Body position (tuck, pike, layout, free)
- Entry type if clear (e.g., vertical entry, feet-first)
- Platform or springboard (if inferable), e.g., 10m platform, 3m springboard
Only output a **single-line caption**, no bullets, no extra explanation.
Format:
[Takeoff type], [# somersaults] somersaults, [# twists if any] twists, [body position].
(Platform: [10m / 3m / Unknown])
Examples:
- Backward takeoff, 2.5 somersaults, tuck. (Platform: 10m)
- Reverse takeoff, 1.5 somersaults, 1 twist, pike. (Platform: 3m)
- Armstand, 2.5 somersaults, layout. (Platform: Unknown)

**Caption Generation Template for LaViLa** As an alternative to VideoXL, we also experimented with LaViLa [7], a general-purpose multimodal model, to generate captions for both query and target videos. We selected LaViLa based on its prior application in EgoCVR [1], a task closely related to CoVR. However, the captions produced by LaViLa lacked the fine-grained detail and domain-specific terminology needed to accurately describe gymnastics and diving routines. This gap is illustrated in Table C1 and Table C2, which compare the official label descriptions from *FineGym* [3] and *FineDiving* [6] with captions generated by LaViLa and VideoXL.

Table C1: Comparison between ground-truth action labels from FineGym and the captions generated by LaViLa and VideoXL. The examples illustrate the inability of both models, particularly LaViLa, to capture fine-grained, domain-specific details such as action type, twist count, and apparatus, which are critical for tasks like *TF-CoVR*.

| Ground-Truth Label | LaViLa Caption | VideoXL Caption |
|---|---|---|
| (Vault) round-off, flic-flac with 0.5 turn on, stretched salto forward with 0.5 turn off | #O A man Y walks around the game | Action: Back Handstand, Turns: 2 |
| (Vault) round-off, flic-flac on, stretched salto backward with 1 turn off | #O person X runs on the ground | Action: Flip, Turns: 3 |
| (Floor Exercise) switch leap with 0.5 turn | #O The woman A runs towards the woman Y | Action: Flip on the floor, Turns: 3 |
| (Floor Exercise) switch leap with 1 turn | #O The man Y jumps down from the wall | Action: Handstand walk with hand release, Turns: 3 |
| (Floor Exercise) johnson with additional 0.5 turn | #O The man Y runs towards the man X | Action: Flip, Turns: 0, Action: Dive, Turns: 0 |
| (Floor Exercise) 2 turn in back attitude, knee of free leg at horizontal throughout turn | #O The woman B falls to the floor | Action: Twirl, Turns: 0 |
| (Floor Exercise) 3 turn on one leg, free leg optional below horizontal | #O The woman Y walks away from the woman X | Action: Flip, Turns: 1 |
| (Floor Exercise) salto forward tucked | #O The woman A raises her hands up | Action: Handstand, Turns: 4 |
| (Floor Exercise) salto forward stretched with 1 twist | #O The woman X throws the ball with the tennis | Action: Handstand on Rungs, Turns: 15 |
| (Floor Exercise) salto backward stretched with 3 twist | #O The man Y throws the slate in his right hand to the ground | Action: Jump from Bar, Turns: 2 |

Table C2: Comparison between ground-truth action labels from FineDiving and captions generated by LaViLa and VideoXL. The examples highlight both models' limitations in capturing critical diving-specific details such as somersault count, twist degree, and entry type. While VideoXL occasionally identifies general action categories, it often fails to reflect the structured semantics required for fine-grained tasks like *TF-CoVR*.

| Ground-Truth Label | LaViLa Caption | VideoXL Caption |
|---|---|---|
| Arm.Forward, 2 Soms.Pike, 3.5 Twists | #O The man X jumps down from the playground slide | Action: Diving, Backflip, Half Turn, T-Walk, Kick flip, Headstand, Handstand, Turns: 3 |
| Arm.Back, 1.5 Twists, 2 Soms.Pike, Entry | #O The girl X jumps down from the playhouse | Action: Flip, Turns: 2 |
| Arm.Back, 2.5 Twists, 2 Soms.Pike, Entry | #O The man X walks down a stair with the rope in his right hand | Action: Gymnasty Turn, Turns: 4 |
| Inward, 3.5 Soms.Pike, Entry | #C C looks at the person in the swimming | Action: Backflip, Turns: 2 |
| Forward, 3.5 Soms.Pike, Entry | #C C shakes his right hand | Action: Dive, Turns: 2 |

Although LaViLa performs well on general video-language benchmarks, it lacks the domain-specific understanding necessary to capture the structured and fine-grained nature of *TF-CoVR* videos. In contrast, targeted prompting with VideoXL produces more consistent and detailed captions, yet it still falls short in accurately identifying the specific actions depicted in *TF-CoVR*.

# D   Experimental Setup

We evaluate *TF-CoVR* using retrieval-specific metrics, namely mean Average Precision at K (*mAP@K*) for $K \in \{5, 10, 25, 50\}$. All models are trained and evaluated on the *TF-CoVR* dataset using varying video-text encoding strategies and fusion mechanisms.

**Video and Text Input Settings.**   We sample 12 uniformly spaced frames from each video and resize them to fit the input dimensions of the pretrained visual backbones. For text input, the modification texts are tokenized using the tokenizer corresponding to each text encoder (e.g., CLIP or BLIP) and passed to the model without truncation whenever possible.

**Text Encoder Evaluation.**   To evaluate the impact of different text encoders on the *TF-CoVR-Base* model, we conducted experiments using two popular pretrained vision-language models: CLIP and BLIP. Both models were used to encode the *modification text* inputs, while the visual backbone and fusion mechanism were held constant (MLP-based fusion with 12-frame video inputs). As shown in Table D3, BLIP consistently outperforms CLIP across all *mAP@K* metrics, suggesting a stronger ability to capture the semantic nuances of the modification texts. Each experiment was repeated five times, and we report the mean and standard deviation to ensure robustness.

Table D3: Evaluation of *TF-CoVR-Base* fine-tuned on *TF-CoVR* with different text encoders using mAP@K for $K \in \{5, 10, 25, 50\}$. We ran each experiment five times and report mean and standard deviation in the following table

| Modalities | | Model | Text | Fusion | #Query | #Target | mAP@K (↑) | | | |
|---|---|---|---|---|---|---|---|---|---|---|
| Video | Text | | Encoder | | Frames | Frames | 5 | 10 | 25 | 50 |
| ✓ | ✓ | TF-CoVR-Base | CLIP | MLP | 12 | 12 | $18.30 \pm 0.35$ | $20.59 \pm 0.30$ | $22.89 \pm 0.27$ | $23.64 \pm 0.27$ |
| ✓ | ✓ | TF-CoVR-Base | BLIP | MLP | 12 | 12 | $20.62 \pm 0.25$ | $23.17 \pm 0.34$ | $25.17 \pm 0.28$ | $25.88 \pm 0.25$ |

**Fusion Module.**   We use a lightweight multi-layer perceptron (MLP) with two hidden layers and ReLU activation to combine visual and textual features, enabling efficient multimodal fusion while preserving architectural simplicity.

**Training and Evaluation Protocols.**   We fine-tune each model using the AdamW optimizer with a learning rate of $1 \times 10^{-4}$ and a batch size of 512. Each model is trained for 100 epochs. All configurations are evaluated across five random seeds to ensure statistical reliability.

**Hardware Configuration and Training Time.**   All experiments were conducted on four NVIDIA A100 GPUs, each with 80 GB of memory. Stage 1 pretraining, performed on two datasets using a single A100 GPU, takes approximately four days, while Stage 2 fine-tuning completes in about six hours.

# E   TF-CoVR Visualization

*TF-CoVR* (Figure E4) offers a clear, structured visualization of the Composed Video Retrieval (CoVR) task, specifically designed for fine-grained temporal understanding. Unlike prior CoVR benchmarks such as WebVid CoVR [5] and EgoCVR [1], which often rely on broad scene-level changes or object variations, *TF-CoVR* centers on subtle, motion-centric transformations. These include variations in the number of turns, transitions between salto types (e.g., *tucked*, *piked*, or *stretched*), and the inclusion or omission of rotational components in gymnastic leaps.

Each row in the figure illustrates a triplet: the left column displays the *query video*, the right shows the corresponding *target video*, and the center presents the *modification text* describing the transformation required to reach the target. *TF-CoVR* emphasizes action-specific, apparatus-consistent changes, where even subtle variations in movement or rotation denote semantically distinct actions. By controlling for background and scene context, the figure isolates fine-grained motion differences as the primary signal for retrieval. This makes *TF-CoVR* a strong benchmark for assessing whether models can accurately retrieve videos based on instruction-driven, temporally grounded modifications. Additional visualizations of *TF-CoVR* are provided in Figures E5 and E6.

## F   Institutional Review Board (IRB) Approval

*TF-CoVR* uses publicly available videos from the *FineGym* and *FineDiving* datasets. Access to these videos is subject to the licensing terms specified by the respective dataset providers. To support reproducibility, we released the video and text embeddings generated during our experiments.

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

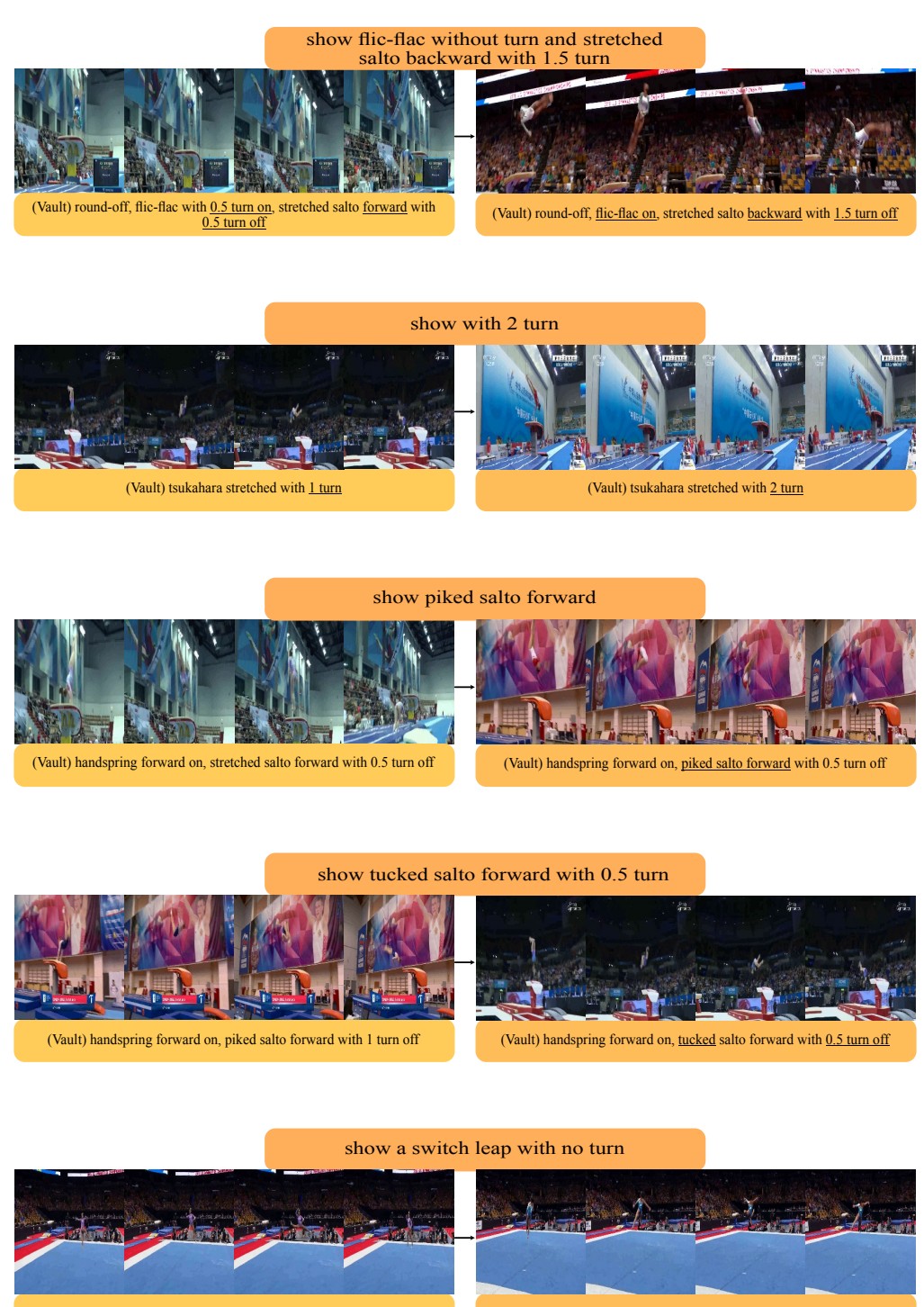

Figure E4: Qualitative examples from *TF-CoVR* showcasing motion-centric transformations for fine-grained temporal action retrieval. The examples span diverse gymnastic events such as *vaults* and *floor exercises*, where subtle differences in execution such as changing from a *stretched* to a *tucked salto*, increasing the number of turns from *one* to *two*, or removing rotation in a *switch leap* define the compositional shift. The captions explicitly highlight these movement attributes, enabling precise instruction-based retrieval grounded in temporal dynamics rather than visual appearance or scene context. This focus on action semantics and minimal visual distraction distinguishes *TF-CoVR* from prior CoVR datasets.

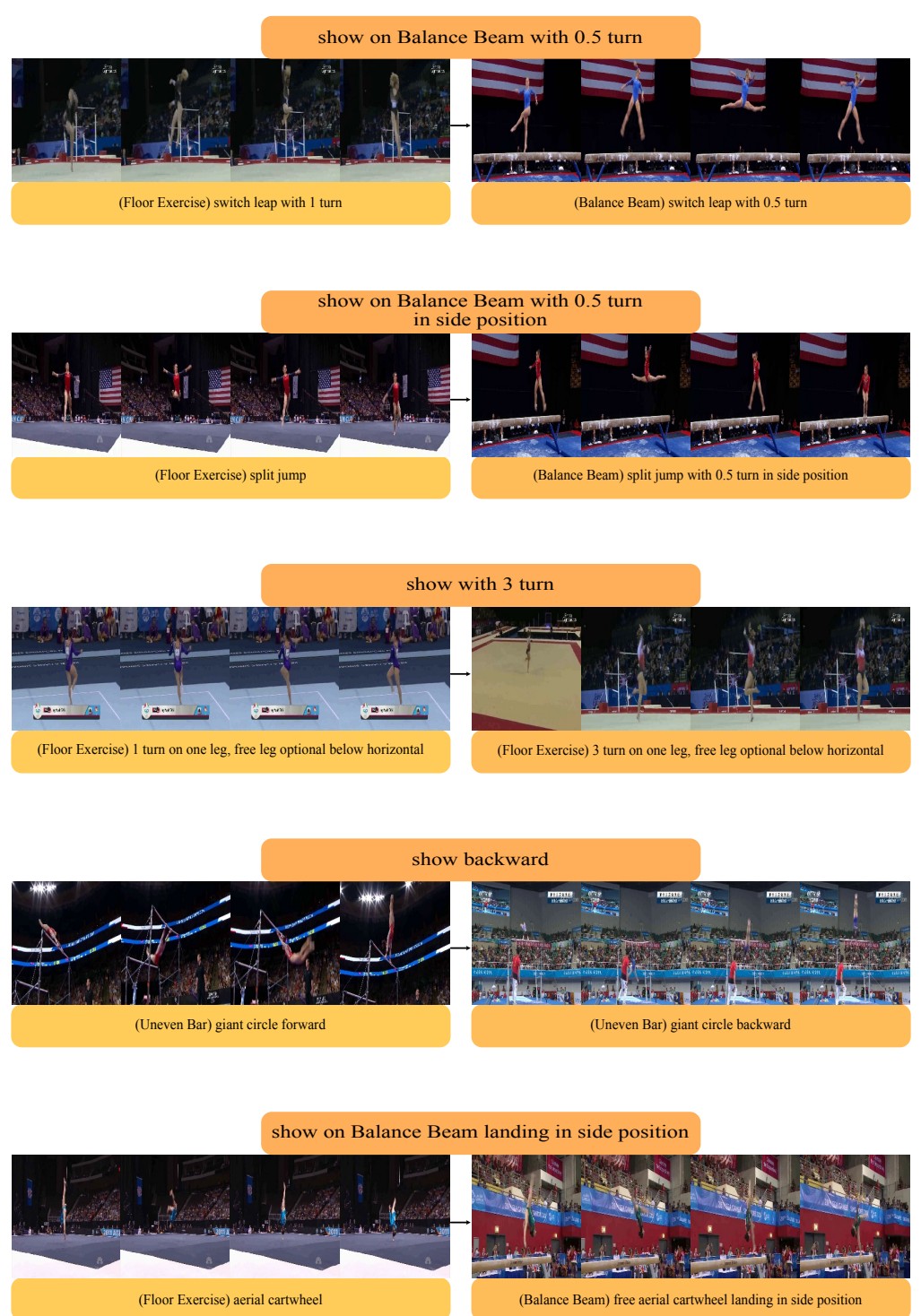

Figure E5: Additional examples from *TF-CoVR* demonstrating temporally grounded modifications across multiple apparatuses. Each triplet reflects precise motion-based transformations driven by modification instructions, such as "*show with 3 turn*", "*show on Balance Beam with 0.5 turn in side position*", or "*show backward*".

**Show with 4.5 somersaults**

Inward, 3.5 Soms.Tuck, Entry → Inward, 4.5 Soms.Tuck, Entry

**Change direction to inward**

Forward, 3.5 Soms.Pike, Entry → Inward, 3.5 Soms.Pike, Entry

**Change direction to inward and show with 1.5 somersaults**

Forward, 3.5 Soms.Pike, Entry → Inward, 1.5 Soms.Pike, Entry

**Show with 2 twists**

Forward, 2.5 Soms.Pike, 1 Twist, Entry → Forward, 2.5 Soms.Pike, 2 Twists, Entry

**Change direction to Forward**

Inward, 3.5 Soms.Tuck, Entry → Forward, 3.5 Soms.Tuck, Entry

Figure E6: *TF-CoVR* triplets from diving events demonstrating precise compositional modifications based on somersault count, twist count, and direction. Examples include transformations such as "*Show with 4.5 somersaults*," "*Change direction to inward*", "*Change direction to inward and show with 1.5 somersaults*", "*Show with 2 twists*", and "*Change direction to forward*". Each caption specifies critical motion semantics like entry type, direction (*forward* or *inward*), somersault type (*Tuck* or *Pike*), and twist count, enabling controlled retrieval grounded in temporally fine-grained action variations.

| Label 1 | Caption 1 | Label 2 | Caption 2 | Label 3 | Caption 3 |
|---|---|---|---|---|---|
| 0 | (Vault) round-off, flic-flac with 0.5 turn on, stretched salto forward with 1.5 turn off | 1 | (Vault) round-off, flic-flac with 0.5 turn on, stretched salto forward with 0.5 turn off | 2 | (Vault) round-off, flic-flac with 0.5 turn on, stretched salto forward with 1 turn off |
| 3 | (Vault) round-off, flic-flac with 0.5 turn on, stretched salto forward with 2 turn off | 4 | (Vault) round-off, flic-flac with 0.5 turn on, 0.5 turn to piked salto backward off | 5 | (Vault) round-off, flic-flac with 0.5 turn on, piked salto forward with 0.5 turn off |
| 6 | (Vault) round-off, flic-flac with 0.5 turn on, piked salto forward off | 7 | (Vault) round-off, flic-flac with 0.5 turn on, tucked salto forward with 0.5 turn off | 8 | (Vault) round-off, flic-flac with 1 turn on, stretched salto backward with 1 turn off |
| 9 | (Vault) round-off, flic-flac with 1 turn on, piked salto backward off | 10 | (Vault) round-off, flic-flac on, stretched salto backward with 2 turn off | 11 | (Vault) round-off, flic-flac on, stretched salto backward with 1 turn off |
| 12 | (Vault) round-off, flic-flac on, stretched salto backward with 1.5 turn off | 13 | (Vault) round-off, flic-flac on, stretched salto backward with 0.5 turn off | 14 | (Vault) round-off, flic-flac on, stretched salto backward with 2.5 turn off |
| 15 | (Vault) round-off, flic-flac on, stretched salto backward off | 16 | (Vault) round-off, flic-flac on, piked salto backward off | 17 | (Vault) round-off, flic-flac on, tucked salto backward off |
| 18 | (Vault) tsukahara stretched with 2 turn | 19 | (Vault) tsukahara stretched with 1 turn | 20 | (Vault) tsukahara stretched with 1.5 turn |
| 21 | (Vault) tsukahara stretched with 0.5 turn | 22 | (Vault) tsukahara stretched salto | 23 | (Vault) tsukahara stretched without salto |
| 28 | (Vault) tsukahara tucked with 1 turn | 28 | (Vault) handspring forward on, stretched salto forward with 1.5 turn off | 28 | (Vault) handspring forward on, stretched salto forward with 0.5 turn off |
| 29 | (Vault) handspring forward on, stretched salto forward with 1 turn off | 28 | (Vault) handspring forward on, piked salto forward with 0.5 turn off | 31 | (Vault) handspring forward on, piked salto forward with 1 turn off |
| 32 | (Vault) handspring forward on, piked salto forward off | 33 | (Vault) handspring forward on, tucked salto forward with 0.5 turn off | 34 | (Vault) handspring forward on, tucked salto forward with 1 turn off |
| 35 | (Vault) handspring forward on, tucked double salto forward off | 36 | (Vault) handspring forward on, tucked salto forward off | 37 | (Vault) handspring forward on, 1.5 turn off |
| 38 | (Vault) handspring forward on, 1 turn off | 40 | (Floor Exercise) switch leap with 0.5 turn | 41 | (Floor Exercise) switch leap with 1 turn |
| 42 | (Floor Exercise) split leap with 0.5 turn | 43 | (Floor Exercise) split leap with 1 turn | 44 | (Floor Exercise) split leap with 1.5 turn or more |
| 45 | (Floor Exercise) switch leap | 46 | (Floor Exercise) split leap forward | 47 | (Floor Exercise) split jump with 1 turn |
| 48 | (Floor Exercise) split jump with 0.5 turn | 49 | (Floor Exercise) split jump with 1.5 turn | 51 | (Floor Exercise) split jump |
| 52 | (Floor Exercise) johnson with additional 0.5 turn | 53 | (Floor Exercise) johnson | 54 | (Floor Exercise) straddle pike or side split jump with 1 turn |
| 55 | (Floor Exercise) straddle pike or side split jump with 0.5 turn | 56 | (Floor Exercise) straddle pike jump or side split jump | 57 | (Floor Exercise) stag ring jump |
| 58 | (Floor Exercise) switch leap to ring position with 1 turn | 59 | (Floor Exercise) switch leap to ring position | 60 | (Floor Exercise) split leap with 1 turn or more to ring position |
| 61 | (Floor Exercise) split ring leap | 62 | (Floor Exercise) ring jump | 63 | (Floor Exercise) split jump with 1 turn or more to ring position |
| 65 | (Floor Exercise) stag jump | 66 | (Floor Exercise) tuck hop or jump with 1 turn | 67 | (Floor Exercise) tuck hop or jump with 2 turn |
| 68 | (Floor Exercise) stretched hop or jump with 1 turn | 69 | (Floor Exercise) pike jump with 1 turn | 70 | (Floor Exercise) sheep jump |
| 71 | (Floor Exercise) wolf hop or jump with 1 turn | 73 | (Floor Exercise) wolf hop or jump | 76 | (Floor Exercise) cat leap |
| 77 | (Floor Exercise) hop with 0.5 turn free leg extended above horizontal throughout | 78 | (Floor Exercise) hop with 1 turn free leg extended above horizontal throughout | 81 | (Floor Exercise) 3 turn with free leg held upward in 180 split position throughout turn |

| Label 1 | Caption 1 | Label 2 | Caption 2 | Label 3 | Caption 3 |
|---|---|---|---|---|---|
| 82 | (Floor Exercise) 2 turn with free leg held upward in 180 split position throughout turn | 83 | (Floor Exercise) 1 turn with free leg held upward in 180 split position throughout turn | 84 | (Floor Exercise) 3 turn in tuck stand on one leg, free leg straight throughout turn |
| 85 | (Floor Exercise) 2 turn in tuck stand on one leg, free leg straight throughout turn | 86 | (Floor Exercise) 1 turn in tuck stand on one leg, free leg optional | 88 | (Floor Exercise) 2 turn in back attitude, knee of free leg at horizontal throughout turn |
| 89 | (Floor Exercise) 1 turn in back attitude, knee of free leg at horizontal throughout turn | 90 | (Floor Exercise) 4 turn on one leg, free leg optional below horizontal | 91 | (Floor Exercise) 3 turn on one leg, free leg optional below horizontal |
| 92 | (Floor Exercise) 2 turn on one leg, free leg optional below horizontal | 93 | (Floor Exercise) 1 turn on one leg, free leg optional below horizontal | 94 | (Floor Exercise) 2 turn or more with heel of free leg forward at horizontal throughout turn |
| 95 | (Floor Exercise) 1 turn with heel of free leg forward at horizontal throughout turn | 97 | (Floor Exercise) aerial cartwheel | 98 | (Floor Exercise) arabian double salto tucked |
| 99 | (Floor Exercise) double salto forward tucked with 0.5 twist | 100 | (Floor Exercise) double salto forward tucked | 101 | (Floor Exercise) salto forward tucked |
| 102 | (Floor Exercise) arabian double salto piked | 105 | (Floor Exercise) double salto forward piked | 104 | (Floor Exercise) salto forward piked |
| 105 | (Floor Exercise) aerial walkover forward | 106 | (Floor Exercise) salto forward stretched with 2 twist | 107 | (Floor Exercise) salto forward stretched with 1 twist |
| 108 | (Floor Exercise) salto forward stretched with 1.5 twist | 109 | (Floor Exercise) salto forward stretched with 0.5 twist | 110 | (Floor Exercise) salto forward stretched, feet land successively |
| 111 | (Floor Exercise) salto forward stretched, feet land together | 112 | (Floor Exercise) double salto backward stretched with 2 twist | 113 | (Floor Exercise) double salto backward stretched with 1 twist |
| 114 | (Floor Exercise) double salto backward stretched with 0.5 twist | 115 | (Floor Exercise) double salto backward stretched | 116 | (Floor Exercise) salto backward stretched with 3 twist |
| 117 | (Floor Exercise) salto backward stretched with 2 twist | 118 | (Floor Exercise) salto backward stretched with 1 twist | 119 | (Floor Exercise) salto backward stretched |
| 120 | (Floor Exercise) salto backward stretched with 3.5 twist | 121 | (Floor Exercise) salto backward stretched with 2.5 twist | 122 | (Floor Exercise) salto backward stretched with 1.5 twist |
| 123 | (Floor Exercise) salto backward stretched with 0.5 twist | 124 | (Floor Exercise) double salto backward tucked with 2 twist | 128 | (Floor Exercise) double salto backward tucked with 1 twist |
| 126 | (Floor Exercise) double salto backward tucked | 128 | (Floor Exercise) salto backward tucked | 128 | (Floor Exercise) double salto backward piked with 1 twist |
| 129 | (Floor Exercise) double salto backward piked | 133 | (Balance Beam) split jump with 0.5 turn in side position | 134 | (Balance Beam) split jump with 0.5 turn |
| 135 | (Balance Beam) split jump with 1 turn | 136 | (Balance Beam) split jump | 137 | (Balance Beam) straddle pike jump with 0.5 turn in side position |
| 138 | (Balance Beam) straddle pike jump with 0.5 turn | 139 | (Balance Beam) straddle pike jump with 1 turn | 140 | (Balance Beam) straddle pike jump or side split jump in side position |
| 141 | (Balance Beam) straddle pike jump or side split jump | 142 | (Balance Beam) stag-ring jump | 143 | (Balance Beam) ring jump |
| 144 | (Balance Beam) split ring jump | 145 | (Balance Beam) switch leap with 0.5 turn | 146 | (Balance Beam) switch leap with 1 turn |
| 147 | (Balance Beam) split leap with 1 turn | 148 | (Balance Beam) switch leap | 150 | (Balance Beam) split leap forward |
| 151 | (Balance Beam) johnson with additional 0.5 turn | 152 | (Balance Beam) johnson | 153 | (Balance Beam) switch leap to ring position |
| 154 | (Balance Beam) split ring leap | 155 | (Balance Beam) tuck hop or jump with 1 turn | 156 | (Balance Beam) tuck hop or jump with 0.5 turn |
| 158 | (Balance Beam) stretched jump/hop with 1 turn | 159 | (Balance Beam) sheep jump | 160 | (Balance Beam) wolf hop or jump with 1 turn |
| 161 | (Balance Beam) wolf hop or jump with 0.5 turn | 162 | (Balance Beam) wolf hop or jump | 163 | (Balance Beam) cat leap |

| Label 1 | Caption 1 | Label 2 | Caption 2 | Label 3 | Caption 3 |
|---|---|---|---|---|---|
| 165 | (Balance Beam) 1.5 turn with free leg held upward in 180 split position throughout turn | 166 | (Balance Beam) 1 turn with free leg held upward in 180 split position throughout turn | 167 | (Balance Beam) 1.5 turn with heel of free leg forward at horizontal throughout turn |
| 168 | (Balance Beam) 2 turn with heel of free leg forward at horizontal throughout turn | 169 | (Balance Beam) 1 turn with heel of free leg forward at horizontal throughout turn | 170 | (Balance Beam) 2 turn on one leg, free leg optional below horizontal |
| 171 | (Balance Beam) 1.5 turn on one leg, free leg optional below horizontal | 172 | (Balance Beam) 1 turn on one leg, free leg optional below horizontal | 173 | (Balance Beam) 1 turn on one leg, thigh of free leg at horizontal, backward upward throughout turn |
| 174 | (Balance Beam) 2.5 turn in tuck stand on one leg, free leg optional | 175 | (Balance Beam) 1.5 turn in tuck stand on one leg, free leg optional | 176 | (Balance Beam) 3 turn in tuck stand on one leg, free leg optional |
| 177 | (Balance Beam) 2 turn in tuck stand on one leg, free leg optional | 178 | (Balance Beam) 1 turn in tuck stand on one leg, free leg optional | 179 | (Balance Beam) jump forward with 0.5 twist and salto backward tucked |
| 180 | (Balance Beam) salto backward tucked with 1 twist | 181 | (Balance Beam) salto backward tucked | 182 | (Balance Beam) salto backward piked |
| 183 | (Balance Beam) gainer salto backward stretched-step out (feet land successively) | 184 | (Balance Beam) salto backward stretched-step out (feet land successively) | 185 | (Balance Beam) salto backward stretched with 1 twist |
| 186 | (Balance Beam) salto backward stretched with legs together | 187 | (Balance Beam) salto sideward tucked with 0.5 turn, take off from one leg to side stand | 188 | (Balance Beam) salto sideward tucked, take off from one leg to side stand |
| 189 | (Balance Beam) free aerial cartwheel landing in side position | 191 | (Balance Beam) free aerial cartwheel landing in cross position | 192 | (Balance Beam) arabian salto tucked |
| 193 | (Balance Beam) salto forward tucked to cross stand | 194 | (Balance Beam) salto forward piked to cross stand | 195 | (Balance Beam) salto forward tucked (take-off from one leg to stand on one or two feet) |
| 196 | (Balance Beam) free aerial walkover forward, landing on one or both feet | 197 | (Balance Beam) flic-flac with 1 twist, swing down to cross straddle sit | 198 | (Balance Beam) flic-flac, swing down to cross straddle sit |
| 207 | (Balance Beam) arabian double salto forward tucked | 208 | (Balance Beam) salto forward tucked with 1 twist | 209 | (Balance Beam) salto forward tucked |
| 210 | (Balance Beam) salto forward piked | 211 | (Balance Beam) salto forward stretched with 1.5 twist | 212 | (Balance Beam) salto forward stretched with 1 twist |
| 213 | (Balance Beam) salto forward stretched | 214 | (Balance Beam) double salto backward tucked with 1 twist | 215 | (Balance Beam) double salto backward tucked |
| 216 | (Balance Beam) salto backward tucked with 1 twist | 217 | (Balance Beam) salto backward tucked | 218 | (Balance Beam) salto backward tucked with 1.5 twist |
| 219 | (Balance Beam) double salto backward piked | 220 | (Balance Beam) salto backward stretched with 3 twist | 221 | (Balance Beam) salto backward stretched with 2 twist |
| 222 | (Balance Beam) salto backward stretched with 1 twist | 223 | (Balance Beam) salto backward stretched | 224 | (Balance Beam) salto backward stretched with 2.5 twist |
| 228 | (Balance Beam) salto backward stretched with 1.5 twist | 226 | (Balance Beam) salto backward stretched with 0.5 twist | 228 | (Balance Beam) gainer salto backward stretched with 1 twist to side of beam |
| 228 | (Balance Beam) gainer salto tucked at end of beam | 229 | (Balance Beam) gainer salto piked at end of beam | 228 | (Balance Beam) gainer salto stretched with 1 twist at end of beam |
| 231 | (Balance Beam) gainer salto stretched with legs together at end of the beam | 232 | (Uneven Bar) pike sole circle backward with 1.5 turn to handstand | 233 | (Uneven Bar) pike sole circle backward with 1 turn to handstand |
| 234 | (Uneven Bar) pike sole circle backward with 0.5 turn to handstand | 235 | (Uneven Bar) pike sole circle backward to handstand | 236 | (Uneven Bar) pike sole circle forward with 0.5 turn to handstand |
| 237 | (Uneven Bar) giant circle backward with 1.5 turn to handstand | 238 | (Uneven Bar) giant circle backward with hop 1 turn to handstand | 239 | (Uneven Bar) giant circle backward with 1 turn to handstand |
| 240 | (Uneven Bar) giant circle backward with 0.5 turn to handstand | 241 | (Uneven Bar) giant circle backward | 242 | (Uneven Bar) giant circle forward with 1 turn on one arm before handstand phase |

| Label 1 | Caption 1 | Label 2 | Caption 2 | Label 3 | Caption 3 |
|---|---|---|---|---|---|
| 243 | (Uneven Bar) giant circle forward with 1 turn to handstand | 244 | (Uneven Bar) giant circle forward with 1.5 turn to handstand | 245 | (Uneven Bar) giant circle forward with 0.5 turn to handstand |
| 246 | (Uneven Bar) giant circle forward | 247 | (Uneven Bar) clear hip circle backward with 1 turn to handstand | 248 | (Uneven Bar) clear hip circle backward with 0.5 turn to handstand |
| 249 | (Uneven Bar) clear hip circle backward to handstand | 280 | (Uneven Bar) clear hip circle forward with 0.5 turn to handstand | 281 | (Uneven Bar) clear hip circle forward to handstand |
| 282 | (Uneven Bar) clear pike circle backward with 1 turn to handstand | 285 | (Uneven Bar) clear pike circle backward with 0.5 turn to handstand | 284 | (Uneven Bar) clear pike circle backward to handstand |
| 285 | (Uneven Bar) clear pike circle forward to handstand | 286 | (Uneven Bar) stalder backward with 1 turn to handstand | 287 | (Uneven Bar) stalder backward with 0.5 turn to handstand |
| 288 | (Uneven Bar) stalder backward to handstand | 289 | (Uneven Bar) stalder forward with 0.5 turn to handstand | 260 | (Uneven Bar) stalder forward to handstand |
| 262 | (Uneven Bar) counter straddle over high bar with 0.5 turn to hang | 263 | (Uneven Bar) counter straddle over high bar to hang | 264 | (Uneven Bar) counter piked over high bar to hang |
| 266 | (Uneven Bar) (swing backward or front support) salto forward straddled to hang on high bar | 267 | (Uneven Bar) (swing backward) salto forward piked to hang on high bar | 268 | (Uneven Bar) (swing forward or hip circle backward) salto backward with 0.5 turn piked to hang on high bar |
| 269 | (Uneven Bar) (swing backward) salto forward stretched to hang on high bar | 280 | (Uneven Bar) (swing forward) salto backward stretched with 0.5 turn to hang on high bar | 281 | (Uneven Bar) transition flight from high bar to low bar |
| 282 | (Uneven Bar) transition flight from low bar to high bar | 285 | (Uneven Bar) (swing forward) double salto backward tucked with 1.5 turn | 284 | (Uneven Bar) (swing forward) salto with 0.5 turn into salto forward tucked |
| 285 | (Uneven Bar) (swing forward) double salto backward tucked with 2 turn | 286 | (Uneven Bar) (swing forward) double salto backward tucked with 1 turn | 287 | (Uneven Bar) (swing forward) double salto backward tucked |
| 288 | (Uneven Bar) (swing backward) double salto forward tucked | 289 | (Uneven Bar) (swing backward) salto forward with 0.5 turn | 280 | (Uneven Bar) (swing backward) double salto forward tucked with 0.5 turn |
| 281 | (Uneven Bar) (under-swing or clear under-swing) salto forward tucked with 0.5 turn | 282 | (Uneven Bar) (swing forward) double salto backward piked | 283 | (Uneven Bar) (swing forward) double salto backward stretched with 2 turn |
| 284 | (Uneven Bar) (swing forward) double salto backward stretched with 1 turn | 285 | (Uneven Bar) (swing forward) double salto backward stretched | 286 | (Uneven Bar) (swing forward) salto backward stretched with 2 turn |
| 287 | (Uneven Bar) (swing forward) salto backward stretched | 407c | Inward, 3.5 Soms.Tuck, Entry | 5285b | Back, 1.5 Twists, 2.5 Soms.Pike, Entry |
| 107b | Forward, 3.5 Soms.Pike, Entry | 6245d | Arm.Back, 2.5 Twists, 2 Soms.Pike, Entry | 207c | Back, 3.5 Soms.Tuck, Entry |
| 5152b | Forward, 2.5 Soms.Pike, 1 Twist, Entry | 5285b | Back, 2.5 Twists, 2.5 Soms.Pike, Entry | 6243d | Arm.Back, 1.5 Twists, 2 Soms.Pike, Entry |
| 109c | Forward, 4.5 Soms.Tuck, Entry | 626c | Arm.Back, 3 Soms.Tuck, Entry | 287c | Reverse, 3.5 Soms.Tuck, Entry |
| 207b | Back, 3.5 Soms.Pike, Entry | 5156b | Forward, 2.5 Soms.Pike, 3 Twists, Entry | 407b | Inward, 3.5 Soms.Pike, Entry |
| 409c | Inward, 4.5 Soms.Tuck, Entry | 6142d | Arm.Forward, 1 Twist, 2 Soms.Pike, 3.5 Twists | 285c | Reverse, 2.5 Soms.Tuck, Entry |
| 405b | Inward, 2.5 Soms.Pike, Entry | 205b | Back, 2.5 Soms.Pike, Entry | 5235d | Back, 2.5 Twists, 1.5 Soms.Pike, Entry |
| 612b | Arm.Forward, 2 Soms.Pike, 3.5 Twists | 105b | Forward, 1.5 Soms.Pike, Entry | 405b | Inward, 1.5 Soms.Pike, Entry |
| 101b | Forward, 0.5 Som.Pike, Entry | 5331d | Reverse, 0.5 Twist, 1.5 Soms.Pike, Entry | 5132d | Forward, 1.5 Soms.Pike, 1 Twist, Entry |
| 614b | Arm.Forward, 2 Soms.Pike, Entry | 5231d | Back, 0.5 Twist, 1.5 Soms.Pike, Entry | 5154b | Forward, 2.5 Soms.Pike, 2 Twists, Entry |

| Label 1 | Caption 1 | Label 2 | Caption 2 | Label 3 | Caption 3 |
|---|---|---|---|---|---|
| 5281b | Back, 1.5 Twists, 2.5 Soms.Pike, Entry | 107c | Forward, 3.5 Soms.Tuck, Entry | 105b | Forward, 2.5 Soms.Pike, Entry |
| 6241b | Forward, 0.5 Twist, 2 Soms.Pike, Entry | 5237d | Back, 3.5 Twists, 1.5 Soms.Pike, Entry | 5353b | Reverse, 1.5 Twists, 2.5 Soms.Pike, Entry |
| 5337d | Reverse, 3.5 Twists, 1.5 Soms.Pike, Entry | 5355b | Reverse, 2.5 Twists, 2.5 Soms.Pike, Entry | 405c | Inward, 2.5 Soms.Tuck, Entry |
| 5335d | Reverse, 2.5 Twists, 1.5 Soms.Pike, Entry | 5172b | Forward, 3.5 Soms.Pike, 1 Twist, Entry | 636c | Arm.Reverse, 3 Soms.Tuck, Entry |
| 205c | Back, 2.5 Soms.Tuck, Entry | 626b | Arm.Back, 3 Soms.Pike, Entry | 401b | Inward, 0.5 Som.Pike, Entry |
| 5233d | Back, 1.5 Twists, 1.5 Soms.Pike, Entry | 109b | Forward, 4.5 Soms.Pike, Entry | 285c | Reverse, 1.5 Soms.Tuck, Entry |