# OpenReview forum: "From Play to Replay: Composed Video Retrieval for Temporally Fine-Grained Videos"
_NeurIPS.cc/2025/Datasets_and_Benchmarks_Track — NeurIPS 2025 Datasets and Benchmarks Track poster_

### Official Review · Reviewer_CAS3 · 2025-06-02

**Ethics Flags:** Human rights (including surveillance)
**Rating:** 5
**Confidence:** 4

**Summary:**

This paper presents TF-CoVR, the first large-scale benchmark for temporally fine-grained composed video retrieval (CoVR). It addresses limitations in existing benchmarks that focus on coarse-grained events or appearance changes. The composed dataset comprises 1.8M triplets from FineGym and FineDiving, with multiple valid target videos per query. Additionally, a baseline method is introduced using a two-stage training pipeline: temporal representation learning followed by contrastive retrieval alignment. The paper benchmarks the standard baseline models (BLIP2, CLIP) , showing some improvements in Zero-shot and fine-tuning regimes. The basic dataset and code are released.

**Additional Feedback:**

- In line 181, 182, the symbol (i and y) definitions are missing. I can easily infer the meaning of i and y but recommend to clarify them for better readability.
- In line 192, BLIPS's reference number is missing.

**Dataset Code Accessibility:**

No

**Dataset Code Comments:**

No problem.

**Ethical Comments:**

There are minor concerns regarding possible visibility of faces in the dataset. The databases are composed with the FineGym and FineDiving, but the authors clarify whether visual privacy was ensured.

**Ethical Considerations:**

No, there are no or only very minor ethics concerns

**Final Justification:**

My concerns are addressed and I move my initial rating to accept.

**Limitations Weaknesses:**

- Rich ablation tests would be helpful for the readers and students. Specifically, the proposed TF-CoVR consists of two stage procedure but there is no detail analysis on the ablation tests on the individual components. For example, how about the performance gain of the stage 2 only without the stage-1?
- In case of zero-shot evalution (Table 2), there are some but limited baselines (e.g., GME, MM, E5) compared with the proposed framework, but in fine-tuned evalution (Table 3), there are only performances of the similar archictures (e.g., BLIPS2, CLIPS). As described in the paper, the proposed method is built on the BLIPS2. It is not fair comparison without other termporal action retrieval methods or CoVR models.
- Using LLM is an interest way but the paper does not check that the generated texts are semantically useful for the task. The best way is to check them by human or checking by BLUE/METEOR metrics. Furthermore, there is no failure case analysis in LLM.
- TF-CoVR is too biased for two sports (FineGym and FineDiving). I just wonder it could be generalized for another domains easily.
- In Table 2, the absolute performance is too low and I wonder it is meaningful for the zero-shot tasks. In this point, I also wonder mAP is a good metric in this case. Why don't use two measurements (e.g., mAP and traditional Recall metric)?
- Is there any way for making end-to-end trainable network architecture?
- Detail analysis on the limitation of CLIP and BLIP is not provided in this paper.

**Strengths Contributions:**

- The main contribution of this paper is to address the first largest-scale benchmark dedicated to temporally fine-grained CoVR. Compared with the previous works, the proposed dataset is designed to introduce a previously unexplored problem setting - temporally fine-grained CoVR. It could be a benchmark set in this field.
- The scale and temporal diversity is larger and richer than prior dataset, specifically, 1.8M triplets, 306 subactions. Furthermore, compared with the previous works, it focuses the temporal fine-grained actions in the sport.
- By using LLM based on label differences, the dataset construction avoids manual caption dependency and has semantically precise, natural-language modifications.
- In Zero-shot evalution, GME, MM, E5 (previous works) are evaluated fairly and the new evaluation metrics (mAP@K) is proposed for checking the performances (but it is also a kind of limitation I think). However, in Fine-tuned evaluation, there is no basic performance comparison with well-known methods.
- The size of the proposed method (1.8M) is large than the previous works in the viewpoint of the fine-grained task, but compared with the general video datasets such as MTCIR, CC-CoIR, it is not significantly larger.

---

> ### Author Rebuttal · Authors · 2025-07-31
>
> - We would like to kindly point the reviewer to *Table 4*, where an ablation of the two-stage framework is already included. Specifically, we report the performance of the model trained with only Stage-2 (without Stage-1 pretraining), which results in a 3.2% drop in mAP@50 (from 25.82 to 22.61). We will revise the table caption to highlight this comparison more clearly and expand the discussion in the text for better visibility.
>
> - Due to the lack of prior CoVR models specifically designed for temporally fine-grained retrieval, we instead benchmarked a diverse set of general multimodal embedding (GME) models (E5-V, MM-Embed, GME-Qwen2-VL) and prior CoVR methods. Interestingly, these models outperformed existing CoVR baselines even in the zero-shot setting, despite not being fine-tuned on our dataset. This shows that strong cross-modal alignment learned from large-scale data can be competitive in fine-grained tasks and highlights the need for future CoVR models that explicitly address temporal structure.
>
> - We manually verified generated modification texts (2413 in total) to ensure semantic correctness, mentioned in L122 in main paper. Additionally, we provided linguistic diversity analysis in the supplementary material (Figure A3) (e.g., distribution of lengths, types of modifiers). We agree that deeper semantic evaluation (e.g., human preference studies, BLEU/METEOR) and failure case analysis are valuable next steps and plan to include them in future iterations.
>
> - Gymnastics and Diving domains were chosen for their rich temporal annotations, but the TF-CoVR framework is domain-agnostic and can be extended to other fine-grained tasks (e.g., cooking, surgical procedures) if such datasets are available.
>
> - TF-CoVR is designed to be a challenging benchmark, as distinguishing fine-grained sub-actions such as 1.5 versus 2 twists requires learning temporally discriminative features. Our zero-shot results exceed strong GME baselines by ~1.6 mAP@50 (7.51 vs. 5.22), and fine-tuned models further close the gap. Regarding metrics, we use mAP@K since multiple correct targets per query make Recall@K less suitable. However, we will consider adding Recall@10 in the appendix.
>
> - An end-to-end trainable architecture is a valuable next step. One possible direction would be to jointly train the temporal representation and the video-text fusion module using a unified loss, allowing the model to directly optimize for retrieval under temporal supervision. This could enable better cross-modal alignment and more flexible adaptation across domains. We note this as an exciting future direction in the conclusion and plan to explore it in follow-up work.
>
> - As discussed in Line 239 of the paper, our proposed two-stage method outperforms existing BLIP-based approaches. This performance gain is attributed to our model’s ability to learn temporally rich features directly from the video input, whereas models like CLIP and BLIP extract features at the frame level and often miss important temporal information. We will expand the discussion in the appendix to clarify how these pre-trained models may struggle to capture subtle temporal cues such as twist count or apparatus variation, which are essential for fine-grained retrieval tasks like TF-CoVR.

---

> > ### Comment · Reviewer_CAS3 · 2025-08-03
> >
> > Thanks for your revision, my major concerns are addressed by your rebuttal. I am happy to move my initial rating to higher one.

---

### Official Review · Reviewer_aw51 · 2025-06-30

**Rating:** 5
**Confidence:** 4

**Summary:**

This paper introduces TF-CoVR, a novel, large-scale dataset for Composed Video Retrieval (CoVR) that targets temporally fine-grained actions in sports videos. The dataset, consisting of 1.8 million video-text triplets from gymnastics and diving domains, is designed to test a model's ability to distinguish subtle differences in motion, such as the number of rotations in a somersault. A key feature of TF-CoVR is its multi-ground-truth evaluation setup, where each query pair has, on average, 3.9 valid target videos, reflecting more realistic retrieval scenarios. The authors constructed the dataset by using the fine-grained action labels from the FineGym and FineDiving datasets and prompting a Large Language Model (LLM) to generate textual modifications describing the differences between pairs of actions.  Alongside the dataset, the paper proposes TF-CoVR-Base, a two-stage baseline model. This model first pre-trains a video encoder on action classification to capture temporal dynamics and then uses a contrastive learning framework to align the composed (video-text) query with target videos. The authors provide a comprehensive benchmark, evaluating existing CoVR methods and General Multimodal Embedding (GME) models, and show that their proposed model achieves state-of-the-art performance on TF-CoVR.

**Additional Feedback:**

Suggestions:
1. The paper would benefit from a more detailed ablation study on the TF-CoVR-Base model. For example, showing the impact of fine-tuning the video encoder in Stage 2, or comparing the MLP fusion with a simple cross-attention mechanism, would provide a more complete picture of the design choices.

Questions:
1. You mention that your modifications "focus exclusively on semantic changes, making them better suited for real-world use cases like highlight generation where visual similarity is not required."  Could you provide examples from your test set where the query and a valid target video are visually very different but semantically linked by the modification text? The qualitative examples in Figure 4 show clips that are quite visually similar.
2. Have you considered using the LLM for more than just generating the modification text? For instance, could it be used in the retrieval model itself to better interpret the compositional query?

**Dataset Code Accessibility:**

Yes

**Dataset Code Comments:**

## Dataset Accessibility, Format, and Documentation
### **Access and Format:**
- **Dataset URL:** [TF-CoVR Dataset on Hugging Face](https://huggingface.co/datasets/animesh007/TF-CoVR)
- **Size:** Approximately 179,000 rows
- **Format:** Parquet
- **Splits:**
  - Train: 178k
  - Validation: 379
  - Test: 473
- **License:** CC-BY-NC-ND 4.0
- **Tools:** Compatible with Hugging Face Datasets and pandas libraries

### **Documentation:**
- The dataset card provides essential metadata, including modalities (text), format (Parquet), size, and licensing information.
- While the dataset card offers a brief overview, it lacks detailed descriptions of the dataset's construction, intended use cases, and potential limitations. ([huggingface.co](https://huggingface.co/datasets/animesh007/TF-CoVR?utm_source=chatgpt.com))

### **Suggestion:**
Enhance the dataset card with comprehensive documentation detailing the dataset's creation process, intended applications, and any known limitations to improve usability and transparency.

---

## Benchmark Reproducibility

### **Motivation and Design:**
- TF-CoVR focuses on temporally fine-grained composed video retrieval, specifically in gymnastics and diving.
- The dataset consists of 180K triplets derived from FineGym and FineDiving datasets.
- Each <query, modification> pair is associated with multiple valid target videos, reflecting real-world tasks such as sports-highlight generation.
### **Evaluation Metrics:**
- The benchmark evaluates models using mean Average Precision at 50 (mAP@50) for zero-shot performance and fine-tuning regimes.

### **Suggestion:**
Provide detailed evaluation protocols, including the specific metrics used, baseline comparisons, and evaluation scripts, to facilitate reproducibility.

---

## Code Accessibility and Reproducibility

### **Repository:**
- **Code URL:** [TF-CoVR Code Repository](https://anonymous.4open.science/r/TF-CoVR-7841/)

### **Contents:**
- The repository includes code for constructing the TF-CoVR dataset and implementing the TF-CoVR-Base framework.
- The code is organized and includes necessary dependencies for running the experiments.

### **Documentation:**
- The repository contains a README file with instructions for setting up the environment and running the code.
- However, the documentation could benefit from more detailed explanations of the code structure, parameter settings, and expected outputs.

### **Suggestion:**
Improve the repository's documentation by providing comprehensive explanations of the codebase, configuration parameters, and step-by-step instructions to enhance reproducibility.

**Ethical Considerations:**

No, there are no or only very minor ethics concerns

**Final Justification:**

The author's rebuttle content has solved most of my concerns. I am happy to raise my rating score!

**Limitations Weaknesses:**

1. Baseline Model Simplicity: The TF-CoVR-Base framework is described as a "concise two-stage training framework."  While effective, its architectural components are relatively standard. The fusion of video and text embeddings is handled by an MLP , which is less complex than the attention-based fusion mechanisms used in other state-of-the-art models. The authors themselves note that an end-to-end, single-stage model might be a "better approach," framing their current model as a strong but potentially preliminary baseline.

2. Domain Specificity: The most significant weakness is the dataset's narrow focus on gymnastics and diving.  While this specialization allows for a deep dive into fine-grained actions, it limits the direct generalizability of models trained on TF-CoVR to other domains. The structured and predictable nature of these sports is not representative of the complexity of fine-grained actions in more varied contexts (e.g., surgical procedures, instructional videos, daily activities). The paper would be stronger if it discussed this limitation more directly and speculated on how these methods might transfer.

3. Limited Analysis of LLM-Generated Text: The paper uses GPT-4 to generate modification texts, but it lacks a deep analysis of the generated language. There is no discussion of the diversity of the generated instructions, their linguistic properties, or potential artifacts and biases that the LLM might introduce. A more thorough characterization of the "modification" part of the triplets would add valuable depth to the dataset's description.

**Strengths Contributions:**

1. Novel and Necessary Benchmark: The primary contribution is the TF-CoVR dataset, which fills a clear and significant gap in the field. Existing CoVR benchmarks often focus on appearance-based changes or coarse-grained events. TF-CoVR is the first to specifically target fine-grained temporal dynamics in fast-paced videos, pushing the community to develop more sophisticated temporal reasoning models. The practical relevance is well-motivated through applications like sports analytics and automated highlight generation.

2. Realistic Multi-Ground-Truth Evaluation: The dataset’s design, which provides multiple valid target videos for a single query, is a major strength.  This structure is more aligned with real-world applications where multiple videos could satisfy a user's request. It also necessitates a more robust evaluation metric (mAP@K), as advocated in the paper, moving beyond the limitations of single-target recall metrics used in many prior works.

3. Comprehensive Benchmarking: The paper provides the first comprehensive study of image, video, and GME models on a temporally fine-grained CoVR task. The inclusion and analysis of GME models in a zero-shot setting is a particularly valuable contribution, revealing that these generalized models are surprisingly competitive with specialized CoVR models on this new task.

4. Presentation and Clarity: The paper is well-written and clearly structured. The motivation for the new dataset is laid out convincingly in the introduction. The figures and tables are highly informative; for instance, Figure 1 effectively visualizes the distinction between TF-CoVR and prior datasets , and Figure 3 offers a clear diagram of the proposed model architecture.

---

> ### Author Rebuttal · Authors · 2025-07-31
>
> - While our method fuses video–text modalities with a simple MLP, exploring more complex attention-based architectures remains an interesting future direction. Our main goal is to provide a concise, reproducible two-stage framework that avoids domain-specific inductive biases, establishing a solid baseline for fine-grained temporal retrieval. We thank the reviewer for highlighting this point and will add, in the paper’s “Future Work” section, plans to investigate attention-based fusion for video–text embeddings.
>
> - Gymnasium and Diving domains were chosen due to the availability of temporally aligned, fine-grained annotations. While our method is domain-agnostic in design, transferring to more varied domains (e.g., surgery, daily tasks) would likely require pretraining on domain-specific datasets.  For example, in surgical setting:
>
>   - The query video might be: Insert needle at a 30-degree angle, advance 2 cm, then begin the suture loop with the right hand.
>
>   - The target video could be: Insert needle at a 45-degree angle, advance 3 cm, then begin the suture loop with the right hand.
>
>   The corresponding modification text would be: “Change needle insertion angle to 45 degrees and advance by 3 cm instead of 2 cm.”
>
>   Understanding such fine-grained, subtle action differences - e.g., variations in motion angle or depth - demands temporally discriminative visual representations aligned with domain-specific text.
>
> - We focused on ensuring the correctness of the generated instructions through manual verification (2413 in total), as mentioned in L122 in paper. We have included an analysis of the linguistic distribution of the modification texts in the supplementary material, please refer to Figure A3. This includes statistics on instruction lengths, and lexical diversity. We agree that deeper exploration of LLM behaviours (e.g., biases, artifacts) would be valuable, and we plan to expand this analysis further in future work.

---

> > ### Comment · Reviewer_aw51 · 2025-08-04
> >
> > Thank to the authors for revision and clarification. The author's rebuttle content has solved most of my concerns. I am happy to raise my rating score!

---

### Official Review · Reviewer_S8NK · 2025-07-03

**Rating:** 4
**Confidence:** 4

**Summary:**

This paper introduces TF-CoVR, the first large-scale benchmark for temporally fine-grained composed video retrieval, constructed from gymnastics (FineGym) and diving (FineDiving) datasets, addressing the limitation of existing benchmarks that only cover appearance changes or coarse event retrieval.

**Dataset Code Accessibility:**

Yes

**Dataset Code Comments:**

This paper provide code and dataset

**Ethical Considerations:**

No, there are no or only very minor ethics concerns

**Final Justification:**

The proposed framework is simple yet effective, including the benchmark dataset. I recommend weak accept.

**Limitations Weaknesses:**

However, there are several issues that I would like the authors to address:
1.The dataset only covers gymnastics and diving. If TF-CoVR-Base is applied to other fine-grained temporal domains (such as dance, cooking or gestures), is domain-specific pre-training required, or can it generalize directly?
2.This paper is currently more application-driven, lacking theoretical analysis and intuitive explanation of why two-stage method is superior to others. If the authors could systematically explain the intrinsic connection between the two-stage training framework and multimodal alignment or temporal modeling, either theoretically or intuitively, it would enhance the depth of the paper
3.I am curious whether the authors have compared HN-NCE with other contrastive loss such as InfoNCE or Triplet Loss? What are the advantages of hard negative weighting in this task?
4.The contribution of Stage-1 pre-training is only implicitly presented in Table 4 through the Stage-2 only comparison, lacking more detailed ablation analysis to quantify its impact.
5.What are the computational costs for Stage 1 and Stage 2 of TF-CoVR-Base?

**Strengths Contributions:**

In terms of methodology, the authors propose TF-CoVR-Base, a two-stage training framework: (i) pre-training a video encoder on fine-grained action classification to obtain temporally discriminative embeddings, and (ii) aligning the composed query (query video + modification text) with candidate videos via contrastive learning. Experiments demonstrate state-of-the-art performance, significantly outperforming existing CoVR and GME baselines. The code are publicly released.
In summary, the new benchmark proposed in this paper reflects real-world applications and help fill a clear research gap in CoVR, while the proposed framework is simple yet effective, as evidenced by extensive experimental results. This is a paper with a good idea and high practical research value.

---

> ### Author Rebuttal · Authors · 2025-07-31
>
> - TF-CoVR-Base currently uses domain-specific datasets for training, but the method itself is domain-agnostic. However, due to the fine-grained and structured nature of different domains (e.g., dance vs. cooking), domain-specific pretraining is helpful to adapt to distinct temporal patterns and visual cues. For example, in the dance domain:
>
>   The query video might be: *Start with right foot step forward, perform moonwalk for 4 counts, then turn left and pose with arms extended.*
>
>   The target video could be: *Start with left foot step forward, perform moonwalk for 2 counts, then turn left and pose with arms extended.*
>
>   The corresponding modification text would be: *“Change starting foot to left foot and moonwalk to 2 counts.”*
>
>   Understanding these subtle temporal variations in action duration and sequencing requires temporally discriminative features. Such granularity is difficult to acquire without appropriate domain-specific pretraining, which helps the model adapt more effectively to the structure of each domain. We will clarify this in the paper and include a discussion on generalization potential and expected challenges.
>
> - Our two-stage design is motivated by the intuition that temporal understanding and multimodal alignment benefit from being decoupled. In Stage 1, the model learns temporally rich video representations via fine-grained action classification, capturing subtle dynamics critical in structured domains like gymnastics. In Stage 2, these pretrained embeddings are aligned with textual modifications using contrastive learning. This separation allows the model to first specialize in temporal modeling, and then focus on semantic grounding across modalities. We would like to highlight our two-stage training is domain agnostic and can be used to all datasets where temporal annotations are available.
>
> - We conducted a comparison between HN-NCE and InfoNCE and observed that InfoNCE performs slightly better achieving 27.13% mAP@50 compared to HN-NCE with 25.82% mAP@50, with approximately 2% higher mAP on our fine-grained video retrieval task. While HN-NCE emphasizes hard negatives through weighting, this strategy may introduce optimization noise in fine-grained settings where many negative samples are highly similar to the positives. In contrast, InfoNCE treats all negatives equally, which can lead to more stable training when subtle visual differences are the key discriminative signals. We will include both results in the final version and update the discussion to reflect this finding. Further, we have shown below the performance comparsion with different values of hard negative weighting:
> | HN-Weighting | mAP@50 |
> |--------------|--------|
> | 0.7          | 23.00  |
> | 0.5          | 25.82  |
> | 0.3          | 26.03  |
>
> - In Table 4, we show the effect of Stage 1 by comparing Stage 2 only (no Stage 1 pretraining) with the full pipeline (Stage 1 + Stage 2), which yields a 3% improvement in mAP@50 (from 22.61 to 25.82). We will clarify this intent in the table caption and can add a supplementary ablation if helpful.
>
> - Stage 1 pretraining (on 2 datasets, 1 GPU (A100 – 80 GB) ) takes ~ 4 days. Stage 2 fine-tuning takes ~ 6 hours. We will include a resource table in the appendix to detail runtime and parameter count.

---

> > ### Comment · Reviewer_S8NK · 2025-08-07
> >
> > The authors have addressed most of my concerns, I keep my original score.

---

### Official Review · Reviewer_w94j · 2025-07-07

**Rating:** 4
**Confidence:** 4

**Summary:**

This paper introduces the TF-CoVR benchmark, focusing on the composed video retrieval (CoVR) issue in the contexts of gymnastics and diving. The primary goal of this benchmark is to evaluate how well models can retrieve target videos based on a query video and a modification text that describes intended changes. The authors also argue that existing CoVR benchmarks emphasize appearance changes and coarse event alterations, which fails to capture fast-paced and subtle temporal differences that are critical for describing sports contexts. To handle this issue, TF-CoVR is presented which collects over 1.8 million triplets set up with a query video, modification text, and target videos. The dataset also includes 306 annotated sub-actions, allowing for comprehensive evaluations.

**Dataset Code Accessibility:**

NA; not applicable to this submission (e.g., no new dataset, benchmark, code, or data provided)

**Ethical Considerations:**

No, there are no or only very minor ethics concerns

**Final Justification:**

Thanks for your reply. I have read the rebuttal, and most of my concerns are solved. Therefore, I keep my positive score.

**Limitations Weaknesses:**

The work focuses on gymnastics and diving scenes only, which limits the generalizability that could benefit from similar fine-grained retrieval tasks. The authors should consider extending the benchmark to various scenes to enhance its generalizability.

The reliance on LLMs for generating modification texts may introduce variability based on the quality of the prompts used. This could affect the consistency of the generated instructions. The authors should consider introducing manual annotation or a check to prove reliability.

**Strengths Contributions:**

It proposes TF-CoVR, which is the first large-scale benchmark designed to fine-grained temporal composed video retrieval, coping with an important problem for the existing works in this area.

The proposed dataset includes 1.8 million triplets with a diverse range of fine-grained actions, making it a representative benchmark for training and evaluating models under dynamic sports environments.

A two-stage training framework, i.e., TF-CoVR-Base is proposed to provide a structured approach to learning temporal context representations. And it can accurately capture subtle action differences in terms of the experiment results.

Through extensive evaluation experiments, the results show clear improvements over previous SOTA models, providing valuable insights into the capabilities of different multimodal embedding models.

---

> ### Author Rebuttal · Authors · 2025-07-31
>
> - While our benchmark currently focuses on FineGym and FineDiving due to the availability of high-quality temporal annotations, our methodology is agnostic to the specific domain. It can be applied to other fine-grained action datasets as they become available. We agree that expanding to additional domains (e.g., cooking, robotics, sports beyond gymnastics) would be a valuable future direction and will release our tools to encourage such extensions.
>
> - To mitigate prompt variability, we manually verified all 2351 modification texts for FineGym and 62 for FineDiving to ensure they accurately reflected the underlying temporal changes, as also mentioned in L122 in paper.
>
>   For example, if the source video label is:
>   “(Vault) round-off, flic-flac with 0.5 turn on, stretched salto forward with 0.5 turn off,”
>
>   and the target label is:
>   “(Vault) round-off, flic-flac on, stretched salto backward with 1 turn off,”
>
>   then a correct modification text should capture the semantic change; e.g.,
>   “show flic-flac on and stretched salto backward with 1 turn.”
>
>   This reflects both the shift in direction (forward → backward) and the update in the salto and turn count. We used this criterion to ensure consistency between the generated text and the actual label transitions.

---

### Decision · Program_Chairs · 2025-09-18

**Decision:**

Accept (poster)

**Comment:**

The article proposes a large-scale benchmark (1.8M triplets) for compositional video retrieval, focusing on gymnastics and diving videos, introducing a baseline for this very fine-grained setting.

All reviewers provided initial positive feedback, highlighting as strengths the fine-grained benchmark (w94j, aw51, CAS3), its scale (CAS3), automatic construction (CAS3), and its realistic challenges (S8NK, aw51). They also appreciated the proposed baseline (w94j, S8NK) and the extensive evaluation (w94j, aw51). They also raised some concerns on the limited focus on gymnastics and diving (w94j, S8NK, aw51, CAS3), ablation of the method's components (S8NK, CAS3), limited baselines (CAS3), and the reliance on LLM for generating modifications (w94j, aw51).

The rebuttal addressed the initial concerns, with all reviewers either confirming or raising their scores. The shared criticisms on the scope were addressed by pointing to the high quality of the collected videos and the possibility of using the same (once released) tools to expand the benchmark.

The AC went to the article, reviews, and rebuttal and agrees with the reviewers. The camera-ready version should include the promised discussion and additional analyses, and the authors are encouraged to release their tools as promised to expand the benchmark further.